# Tradeoff between lag time and growth rate drives the plasmid acquisition cost

Mehrose Ahmad[1], Hannah Prensky[1], Jacqueline Balestrieri[1], Shahd ElNaggar [1], Angela Gomez-Simmonds [2], Anne-Catrin Uhlemann [2], Beth Traxler[3], Abhyudai Singh [4] & Allison J. Lopatkin [1,5,6,7,8] ✉

Conjugative plasmids drive genetic diversity and evolution in microbial populations. Despite their prevalence, plasmids can impose long-term fitness costs on their hosts, altering population structure, growth dynamics, and evolutionary outcomes. In addition to long-term fitness costs, acquiring a new plasmid introduces an immediate, short-term perturbation to the cell. However, due to the transient nature of this plasmid acquisition cost, a quantitative understanding of its physiological manifestations, overall magnitudes, and population-level implications, remains unclear. To address this, here we track growth of single colonies immediately following plasmid acquisition. We find that plasmid acquisition costs are primarily driven by changes in lag time, rather than growth rate, for nearly 60 conditions covering diverse plasmids, selection environments, and clinical strains/species. Surprisingly, for a costly plasmid, clones exhibiting longer lag times also achieve faster recovery growth rates, suggesting an evolutionary tradeoff. Modeling and experiments demonstrate that this tradeoff leads to counterintuitive ecological dynamics, whereby intermediate-cost plasmids outcompete both their low and high-cost counterparts. These results suggest that, unlike fitness costs, plasmid acquisition dynamics are not uniformly driven by minimizing growth disadvantages. Moreover, a lag/growth tradeoff has clear implications in predicting the ecological outcomes and intervention strategies of bacteria undergoing conjugation.

Horizontal gene transfer (HGT), particularly plasmid conjugation, plays a significant role in microbial diversity and evolution[1,2]. Conjugation involves the movement of plasmid DNA from a donor to a recipient through a transmembrane Type IV secretion channel, producing a transconjugant[3,4]. HGT, and especially conjugation, is thought to be frequent in microbial communities, providing opportunistic pathogens and commensal microbes immediate access to diverse genes encoding traits such as virulence factors or metabolic enzymes[5–10]. Indeed, plasmid conjugation is considered the primary way that antibiotic resistance genes spread[11–14]. Thus, it is critical to understand factors facilitating plasmid dissemination to identify strategies to predict, prevent, and/or control such occurrences.

[1]Department of Biology, Barnard College, New York, NY 10027, USA. [2]Department of Medicine, Division of Infectious Diseases, Columbia University Medical Center, New York, NY 10032, USA. [3]Department Microbiology, University of Washington, Seattle, WA 98195, USA. [4]Department of Electrical and Computer Engineering, University of Delaware, Newark, DE 19717, USA. [5]Department Ecology, Evolution, and Environmental Biology, Columbia University, New York, NY 10027, USA. [6]Data Science Institute, Columbia University, New York, NY 10027, USA. [7]Department of Chemical Engineering, University of Rochester, Rochester, NY 14627, USA. [8]Department of Microbiology and Immunology, University of Rochester Medical Center, Rochester, NY 14627, USA. ✉e-mail: allison.lopatkin@rochester.edu

The growth effects of plasmids on their hosts are typically characterized by comparing the relative fitness of an established plasmid-carrying strain to its plasmid-free counterpart[15–17]. This approach has revealed that plasmids are often metabolically burdensome to their hosts, resulting in an observable fitness cost[14,18–21]. Although compensatory processes can alleviate this growth reduction, fitness costs have been shown to dictate the extent and trajectory of plasmid spread in microbial environments[22–24]. Plasmid fitness costs describe growth effects *after* plasmids establish themselves in new host strains; though informative, these measurements do not capture dynamics inherent in initial plasmid establishment.

Separate from a fitness cost, establishing a novel plasmid in a new host introduces a transient metabolic perturbation that is burdensome to the cell. Specifically, immediately following plasmid acquisition, a cell must reallocate intracellular resources to accommodate new energetic requirements associated with plasmid DNA replication and gene expression[25–29]. Consequently, the magnitude and duration of the burden depends on a complex interplay between multiple interdependent biomolecular factors; for example, energy availability at the time of acquisition, plasmid-specific gene regulatory networks, and the translation order of plasmid-encoded genes, may all result in either delayed or over-production of proteins[30]. In addition, acquiring new DNA is known to activate energetically burdensome stress responses, further exacerbating the immediate burden of a new plasmid[31,32]. Combined, these short-term effects may impact fundamental bacterial physiology, clonal and population selection dynamics, and ultimately plasmid dispersal and success[33].

Despite its clear significance, few studies have examined these short-term effects of plasmid acquisition, termed the plasmid acquisition cost[4,30]. In all cases, growth defects were measured in liquid culture, and across few or inconsistent conditions; this limits both the generalizability, as well as the ecological implications, of plasmid acquisition effects. Further, previous population-level studies are unable to distinguish between changes in lag time, growth rate, or both. However, lag times and growth rates each uniquely contribute to various facets of bacterial dynamics, including robustness in fluctuating environments[34], sensitivity to stressors[35], and competitive success[36]. Thus, this distinction is critical to better understand both the physiological manifestation of acquisition costs, and as a corollary, the selection dynamics at play in clonally heterogeneous populations most representative of those in nature[37–41]. Overall, the generality, ecological consequences, and fundamental physiological effects of plasmid acquisition, remain critical open questions.

Here, to address these gaps, we characterize transient plasmid-mediated growth defects at the single-colony level, allowing us to quantify plasmid acquisition costs more rigorously and systematically. We find that the acquisition cost is primarily driven by changes in lag time rather than growth rate across diverse and clinically relevant plasmids, antibiotics, and Gram-negative strains and species. Moreover, our analysis reveals that individual clones carrying higher-cost plasmids counterintuitively exhibit a tradeoff, whereby faster post-lag growth rates temper the potential disadvantages of longer lag times. Finally, we demonstrate the ecological implications of this tradeoff – a potential selection advantage for plasmids with intermediate acquisition costs. Ultimately, our work demonstrates the complex interplay between plasmid acquisition and overall bacterial growth dynamics, and further highlights the critical role of plasmid acquisition in understanding and predicting HGT outcomes in complex microbial communities.

## Results

### Quantifying the plasmid acquisition cost at the single-colony level

During conjugation, a donor transfers a copy of its plasmid to a recipient, generating a de novo transconjugant (Fig. 1A). This de novo transconjugant undergoes a period of adaptation as the plasmid establishes itself in the host (e.g., reaching the appropriate copy number, expressing plasmid-encoded genes). Ultimately, by 24 h (although likely sooner), adaptation is complete, and cells can be considered fully adapted transconjugants[4].

The cost associated with plasmid acquisition can be quantified using a time-to-threshold (TTT) metric. TTT is defined as the time it takes a bacterial population to reach a 'threshold' density in the exponential phase, and captures changes in both lag time and/or growth rate as a result of plasmid acquisition[4]. However, estimating TTT and resulting growth parameters from liquid culture poses two main limitations: these values potentially (1) reflect the extremes of the entire population, which can be driven by a few dominant clones and (2) are confounded by competition amongst individual cells within the population.

To control for these factors, and to quantify the effects of plasmid acquisition more directly, we modified this approach to increase both its quantitative resolution and throughput (Fig. 1B). Briefly, de novo transconjugants are generated in the same way as previously (see Methods)[4]; similarly, adapted transconjugants (i.e., cells that have acquired the plasmid at least 24 h prior) are maintained under identical conditions. Following the conjugation period, both populations are plated onto a dual-antibiotic agar plate such that single transconjugant colonies are visible; dual-antibiotic selection is used to uniquely select for the plasmid-carrying population (e.g., de novo and adapted), while inhibiting the growth of residual parents present in the conjugation mixture. Finally, agar plates are placed onto a temperature-controlled flatbed scanner. Automated images are collected every 15 min over 24 h. This method directly tracks individual de novo and adapted colonies, thereby minimizing any confounding effects of batch population dynamics. Further, since distinct colonies originate from single cells, the growth curves of individual de novo and adapted colonies can be compared directly. The acquisition cost can then be calculated as the ratio between the average TTT of de novo compared to adapted colonies. Finally, since both populations grow on two sides of the same agar plate, this modified protocol also controls for environmental variability.

To confirm the validity and consistency of this scanner-based method, we first verified that neither our setup nor analysis was biased towards colonies growing on either side of the agar plate. We split a population of plasmid-free *Escherichia coli* recipient cells, plating each portion onto different sides of the same plate. Results indicated that TTT from each side of the plate were statistically identical (Supplementary Fig. 1a, $p = 0.812$). We then verified that a plasmid previously identified as costly in liquid culture, RP4, remained costly using this new approach. As expected, our method detected an acquisition cost for RP4 under tetracycline selection: growth curves of de novo cells exhibited a clear growth defect compared to adapted transconjugants (Fig. 1C, left); this significantly increased the TTT for de novo cells (Fig. 1C, right). This finding remained true regardless of the threshold density used to quantify TTT (Supplementary Fig. 1b). Thus, combined, these results demonstrate that our scanner-based approach is both consistent with, and provides greater resolution than, the previous methodology.

### Lag time drives acquisition cost rather than growth rate under tetracycline selection

Aside from validating previous work, colony tracking also revealed extensive clonal heterogeneity within the de novo population; here, we use heterogeneity to denote individual colonies that belong to the same genetically identical population but exhibit phenotypic differences, as evidenced in the variability of growth curves for individual de novo RP4 transconjugants. Since such clonal differences would not be detectable in liquid culture, this heterogeneity had not been previously quantified. Thus, to further investigate this more rigorously, we quantified the growth rate and lag time of RP4 de novo and adapted

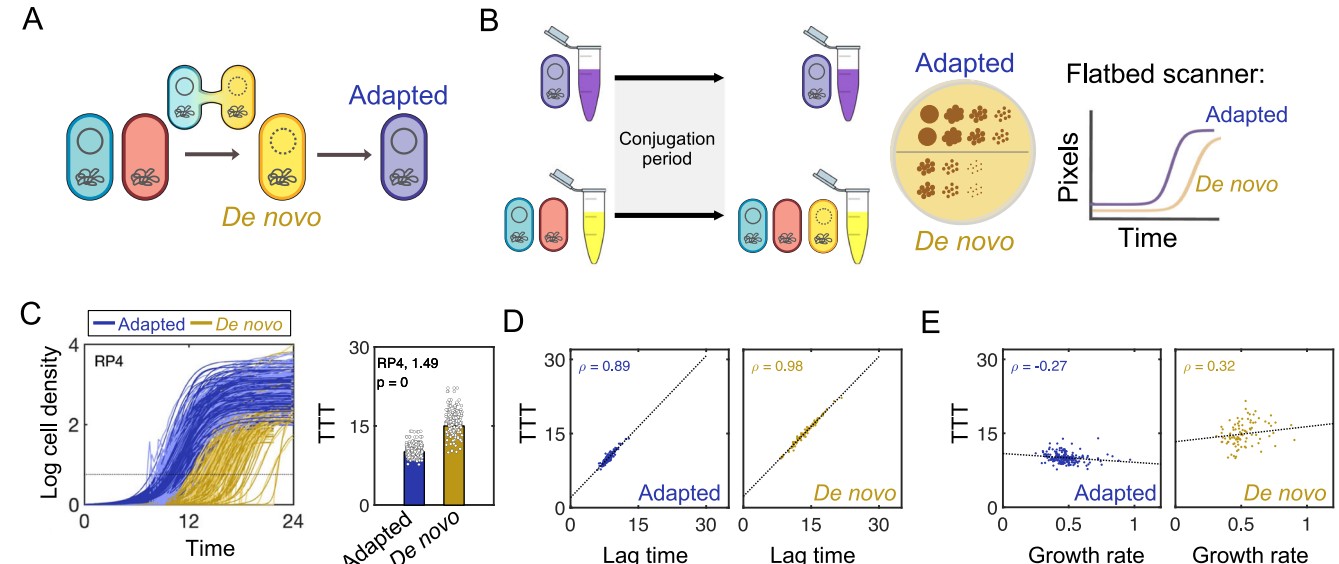

**Fig. 1 | Quantifying plasmid acquisition costs with single-colony resolution.**
**A** Schematic illustration of conjugation between a donor (blue) and recipient (red) cell, generating a de novo transconjugant (yellow). Over time, the de novo transconjugant adapts to the plasmid, becoming an adapted transconjugant (purple). **B** Schematic illustration of the acquisition cost protocol. **C** Left: Log cell density in pixels (y-axis) of RP4 de novo (yellow, $n = 96$) and adapted (purple, $n = 208$) transconjugants over time (x-axis, hours). The black horizontal line indicates the time-to-threshold (TTT) in hours. Right: TTT (y-axis) of de novo (yellow) compared to adapted (purple) transconjugants. Markers correspond to the TTT of individual colonies pooled across seven independent biological replicates. Bar height

represents the mean TTT of all colonies. The acquisition cost (1.49) is determined to be statistically significant using a two-sided $t$-test ($p$ value shown) between the TTT of adapted and de novo populations. **D, E** Individual de novo (yellow, right) and adapted (purple, left) colony lag times (**D**) or growth rates (**E**) (x-axis) plotted against their corresponding TTT (y-axis). In all cases, TTT is defined as the time in hours it takes the bacterial population to reach a 'threshold' density within the exponential phase; the threshold used here is 0.8. Where applicable, the Pearson correlation coefficient is reported ($\rho$) and the linear regression line of best fit is shown in black. All Fig. 1 data uses recipient strain RB933. All Source data are provided as a Source Data file.

transconjugant colonies. For both population types, lag times of individual colonies increased linearly with TTT; that is, de novo and adapted clones with longer lag times took longer to reach the threshold density (Fig. 1D). Moreover, adapted transconjugants exhibited an inverse relationship between growth rate and TTT, where adapted clones growing faster reached the threshold density sooner (Fig. 1E, left). However, de novo transconjugants did not follow this trend: the growth rate of these cells showed a slight positive correlation with TTT (Fig. 1E, right, $p < 0.005$). Initially, this suggested that following RP4 acquisition, acquisition cost dynamics were not significantly impacted by growth rates, and instead were primarily driven by lag times.

To determine whether this trend was plasmid-specific, we next quantified acquisition costs for eight additional plasmids under identical antibiotic selection and recipient strain conditions (Fig. 2A, top row); this initial focus on the bacteriostatic drug tetracycline is ideal, since it minimizes potential metabolic effects imposed by bactericidal antibiotics that may confound lag/growth dynamics[42–44]. These plasmids ranged in size, copy number, isolation source (i.e., clinical or environmental), and incompatibility (Inc) group (IncI, F, P, and N) (Supplementary Table 1). Quantifying lag time and growth rates for all nine plasmids indicated that results were consistent with RP4: single colonies of de novo and adapted transconjugants exhibited the same direct, proportional trend between lag time and TTT (Fig. 2A, bottom row). Moreover, while adapted cells retained the inverse correlation between growth rate and TTT, de novo cells exhibited an apparent decoupling between growth rate and TTT for costlier plasmids (Fig. 2A, top and middle rows). Plasmids with no or low acquisition costs (e.g., R100-1, R64drd, and pOX38) retained the inverse relationship between growth rate and TTT, whereas this inverse relationship was lost for high-cost plasmids (pB10 and pRK100).

Pooling together all 9 plasmids revealed that the average plasmid acquisition cost in de novo transconjugants was linearly correlated

with the average lag time (Fig. 2B, left, $p = 8.33 \times 10^{-5}$). However, the average growth rate in de novo populations exhibited no significant relationship with acquisition cost (Fig. 2C, left, $p = 0.23$). In contrast, for adapted transconjugants, there was no significant relationship between the acquisition cost and either lag time or growth rate (Fig. 2B, C, right panels, $p > 0.1$). This finding was expected: since adapted cells no longer experience the physiological effects of plasmid acquisition, any relationship between the acquisition cost and growth effects was anticipated to be de novo-specific.

We next tested whether these results were specific to our quantification methods. First, we examined whether the threshold density chosen for TTT impacts the relative importance of lag time and growth rate on the acquisition cost. For all threshold densities reasonably spanning exponential phase, lag time consistently and strongly correlated with acquisition cost, while growth rate did not (Supplementary Fig. 2a). Next, we verified whether these results were specific to the Logistic-curve-fitting method used to quantify growth rates. Indeed, we observed identical trends when using an alternative fitting procedure (See Methods/Supplement) (Supplementary Fig. 3a, b)[45]. Moreover, instantaneous colony growth rates (e.g., the change in colony size at every time point as a function of the overall average colony size) further validated these results: despite differences in colony sizes, de novo colonies ultimately attained the same growth rates as adapted colonies, suggesting that overall trends are not driven by size-specific artifacts of curve fitting (Supplementary Fig. 3c). Combined, these results confirm that under tetracycline selection, acquisition costs are driven by changes in lag time, not growth rate.

## Lag time drives acquisition cost for diverse conditions and is independent of plasmid-specific features

To further examine the generality of this observation, we first investigated whether any common plasmid features could readily explain these results. Specifically, we examined the relationship between the

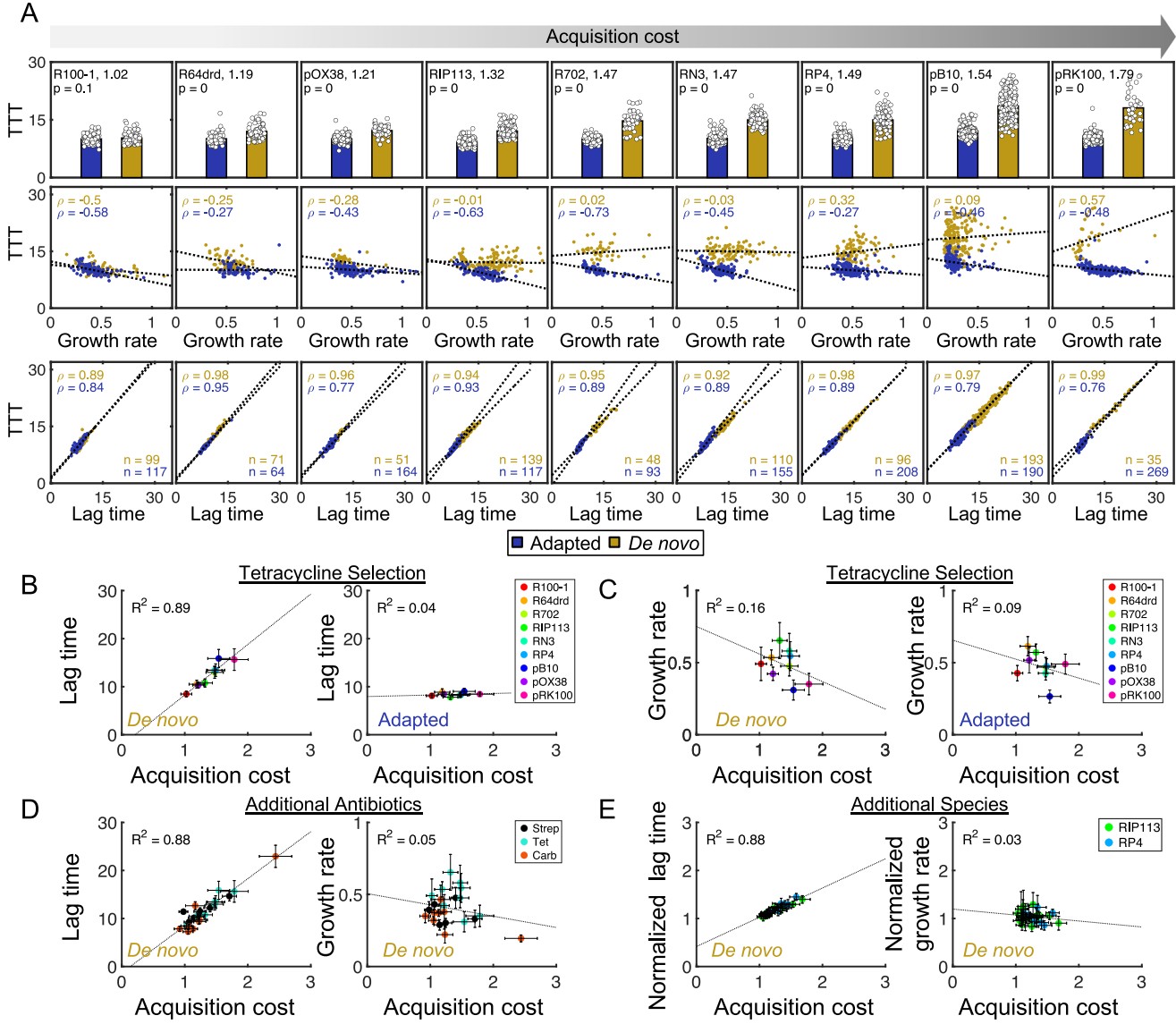

**Fig. 2 | Lag time, not growth rate, drives plasmid acquisition cost. A** Top: De novo (yellow) and adapted (purple) transconjugants TTT in hours (hr) for nine plasmids under tetracycline/rifampicin selection. The top left values correspond to acquisition cost magnitude (i.e., 1.02 for R100−1). Bar height represents the mean TTT of all *n* colonies. Middle: Growth rate (hr$^{-1}$) versus TTT. Bottom: Lag time (hr) versus TTT. Markers represent individual colonies. *P* values in the top left corner for each plasmid are calculated from Bonferroni-corrected two-sided *t*-tests. The number of colonies (*n*) per population is displayed in the bottom row each panel. Plasmids increase in acquisition cost from left to right (gray arrow). **B**, **C** Lag time (y-axis) (**B**) or growth rate (**C**) for all nine plasmids from (**2A**) plotted as a function of acquisition cost (x-axis) for both de novo (left) and adapted (right) transconjugants. Individual markers correspond to distinct plasmids: R100-1 (red), R64drd (orange), R702 (chartreuse) RIP113 (green), RN3 (mint), RP4 (blue), pB10 (dark blue), pOX38 (purple), and pRK100 (pink). Data for **A**–**C** are pooled across at least three independent biological replicates, and the resulting total number of colonies (*n*) per population is shown in the bottom row

of **2A** (i.e., *n* = 99 and 117 for R100-1 de novo and adapted colonies, respectively); these numbers indicate the sample sizes for statistics. **D** Lag time (left) or growth rate (right) of de novo transconjugants plotted as a function of acquisition cost for eleven additional plasmids under either carbenicillin (orange) or streptomycin (black) selection; data is combined with tetracycline (blue) data (i.e., from **2A**, **B**). Individual markers correspond to mean values for distinct plasmids. Data in **2A**–**D** use RB933 recipient. **E** Lag time (left) or growth rate (right), each normalized to the corresponding recipient's adaptive values, plotted as a function of acquisition cost for RIP113 (green) and RP4 (blue), each with 24 or 7 unique *Klebsiella pneumonia* strains, respectively. For **2D**, **E**, colonies were pooled across at least two independent biological replicates (see Supplementary Table 1c for sample sizes and replicate numbers). In all cases, individual markers represent means and error bars represent standard deviations across all *n* colonies. Black lines represent the linear regression line of best fit, and either the Pearson correlation coefficient (ρ) or $R^2$ values are reported. All Source data are provided as a Source Data file.

acquisition cost and the number of transconjugants (as a proxy for the conjugation efficiency), the adapted transconjugant growth rate (as a proxy for fitness cost), and the overall plasmid size. In all cases, plasmid features were not significantly correlated with the plasmid acquisition cost (Supplementary Fig. 2b, *p* > 0.5). Likewise, genetic features related to tetracycline were also not predictive of acquisition cost. Specifically, neither the tetracycline operon sequence/

organization, nor the location of *tetA* on the plasmid relative to the origin of transfer (*oriT*) or replication (*oriC*), were correlated with the acquisition cost (Supplementary Fig. 2c and Supplementary Table 2). Combined, this suggests that lag time trends are independent of general plasmid, and tetracycline-specific sequence, features.

Having ruled out plasmid- or tetracycline-specific genetic explanations, we next tested whether these results were specific to our

initial choice of donor (tetracycline) and recipient (rifampicin) selection schemes. Since rifampicin resistance is expressed in recipients even before plasmid acquisition, we reasoned this drug likely would not impact the adaptation process. To test this, we used kanamycin instead of rifampicin to select the same recipient, and measured the acquisition cost for both R100-1 and RIP113 plasmids. For both plasmids, the magnitude of the acquisition cost, the statistically significant relationship between lag time and TTT, and the statistically insignificant relationship between growth rate and TTT, were maintained (Supplementary Fig. 4). After verifying that recipient selection did not influence our results, we next focused on resistance specified by the plasmid itself. Here, because the recipient has never been exposed to tetracycline prior to transconjugant selection, it would be reasonable to expect that the observed decoupling may be driven by tetracycline-specific effects. To test this potential specificity, we measured acquisition costs for 11 additional plasmids (14 new conditions overall) under rifampicin recipient selection and either streptomycin or carbenicillin donor selection (Supplementary Table 1); these two antibiotics were chosen due to their distinct mechanisms of action, enzyme-based resistances, and potential for metabolic dysregulation, as compared to tetracycline[46–48]. For all plasmids/drugs, as with tetracycline, acquisition cost and lag time showed a strong positive and statistically significant relationship, whereas growth rate remained uncorrelated with acquisition cost (Fig. 2D). Combined, this confirmed the observed trends are general to diverse bactericidal and bacteriostatic antibiotics.

Finally, we wondered whether our results were specific to *E. coli* as the recipient strain. To test this, we used a panel of 24 rifampicin-resistant *Klebsiella pneumoniae* (KPN) clinical isolates and quantified the acquisition cost using the IncN plasmid RIP113 and the IncP plasmid RP4. Even in this additional species, lag time, and not growth rate, correlated with acquisition cost (Fig. 2E). Overall, these results conclusively demonstrate that the acquisition cost is primarily driven by lag time, rather than the growth rate.

### Tradeoff between lag time and growth rate underlies costly plasmid acquisition

Despite the primary role of lag time in driving the acquisition cost, the negative correlation between growth rate and TTT observed with low acquisition cost plasmids appeared to become decoupled, and even positively correlated, as the acquisition cost increased. In other words, for a costly plasmid, individual clones that took longer to reach TTT exhibited a counterintuitively higher growth rate; this suggested that growing too early was overall detrimental to plasmid establishment. On the other hand, longer lag times, while otherwise disadvantageous, seemed to be tolerable, as they conferred clones a relative growth advantage. To investigate this further, we examined the relationship between lag time and growth rate for all 9 plasmids quantified under tetracycline selection. Consistent with TTT results, lag time and growth rate became positively correlated for sufficiently costly plasmids; the tradeoff between lag time and growth rate appeared to increase with acquisition cost (Fig. 3A).

Given observed clonal variability in lag times and growth rates, we first sought to rule out the possibility that the counterintuitive relationship between these parameters arose by chance as an artifact of noise. To do so, we performed lognormal bootstrapping using distributions parametrized by means and standard deviations of de novo lag times calculated from experimental data. Specifically, for each plasmid, we generated 300 independent random lag time and growth rate pairs under two different hypotheses (see Supplementary Methods). In the first hypothesis, lag time and growth rate were uncorrelated to each other; this corresponded to the assumption that the observed positive relationship between lag time and growth rate occurs by chance. In the second, lag time was constrained by growth rate. Under both scenarios, we used each of the 300 pairs to calculate a

logistic growth curve; from this growth curve, we then determined both TTT, and the dependence between growth rate and TTT. This process was repeated 1000 independent times, yielding a distribution of growth rate/TTT dependences under both hypotheses. Results indicated that only when lag time-constrained growth rates, the latter's correspondence to TTT could be accurately predicted. In other words, a constrained model could predict the observed decoupling between growth rate and TTT, as judged by overlapping 95% confidence intervals of the predicted and observed values across all plasmids tested (Fig. 3B, top row). Conversely, the model without a lag time/growth rate constraint could not predict TTT dependence on growth within a 95% confidence interval (Fig. 3B, bottom row). Together, these results suggest that the observed decoupling between growth rate and TTT in costly plasmids is driven by a counterintuitive direct relationship between lag time and growth rate: as lag time increases, so does growth rate.

Combined, our results confirm that the tradeoff between lag time and growth rate is a feature of the acquisition cost and likely not an artifact of the data. Thus, to quantify this tradeoff, we calculated the slope of the regression line between growth rate and lag time for each of the 23 plasmids/conditions measured using the *E. coli* recipient. Pooling all data showed that, consistent with our interpretation, the slope (i.e., the strength of the relationship) was positively correlated with acquisition cost more than would be expected by chance alone (Fig. 4A, $\rho = 0.91$); this was also true for the acquisition cost of *K. pneumoniae* strains with both RP4 and RIP113 plasmids (Supplementary Fig. 5, $\rho = 0.66$).

### Ecological implications favor intermediate acquisition costs

Our data reveals two key trends: (1) the adaptation period following plasmid acquisition primarily manifests in a prolonged lag time and (2) this prolonged lag time dictates the resulting growth rate of transconjugants—cells with longer lag times (i.e., slower adaptation) enjoy faster growth rates, and these growth advantages increase with the plasmid acquisition cost. During the lag phase, cells metabolically adapt to optimize growth in that specific environment[49]. Given that all co-existing cells must both adapt and utilize a shared resource pool simultaneously, these two trends combined likely impact both intra and interpopulation competition dynamics. Thus, we sought to assess how the lag/growth tradeoff, which is inherently dependent on underlying clonal heterogeneity, would impact selection dynamics and ecological outcomes.

We first examined selection dynamics at the intrapopulation level; that is, how does the lag/growth tradeoff impact clonal selection within a genetically identical, single-plasmid population? Our data suggests that for a costly plasmid, cells taking longer to adapt are more likely to attain faster growth rates. However, the longer cells take to begin growing, the less competitive they will be, due to resource depletion and carrying capacity constraints. Thus, increased competition amongst individual clones will reduce the effective growth advantage for cells with a longer lag time. However, individual cells that initiate growth relatively quickly may still be outcompeted by those with a longer lag time, so long as the latter's growth rate is fast enough to compensate. To test this idea, we varied the initial cell density of a single-plasmid population (either de novo or adapted) to simulate increasing clonal heterogeneity. We used a simplified ordinary differential equation model consisting of a population (N) of *J* unique clones. In each case, clonal lag times were drawn from experimentally-determined distributions, and corresponding growth rates were calculated using the tradeoff relationship (or lack thereof) inferred from representative plasmids (Supplementary Fig. 6a). For all simulations, 50 iterations (i.e., unique parameter distributions) were averaged to ensure that predictions reflected average dynamics rather than artifacts of numerically-generated distributions of varying sizes/values. Finally,

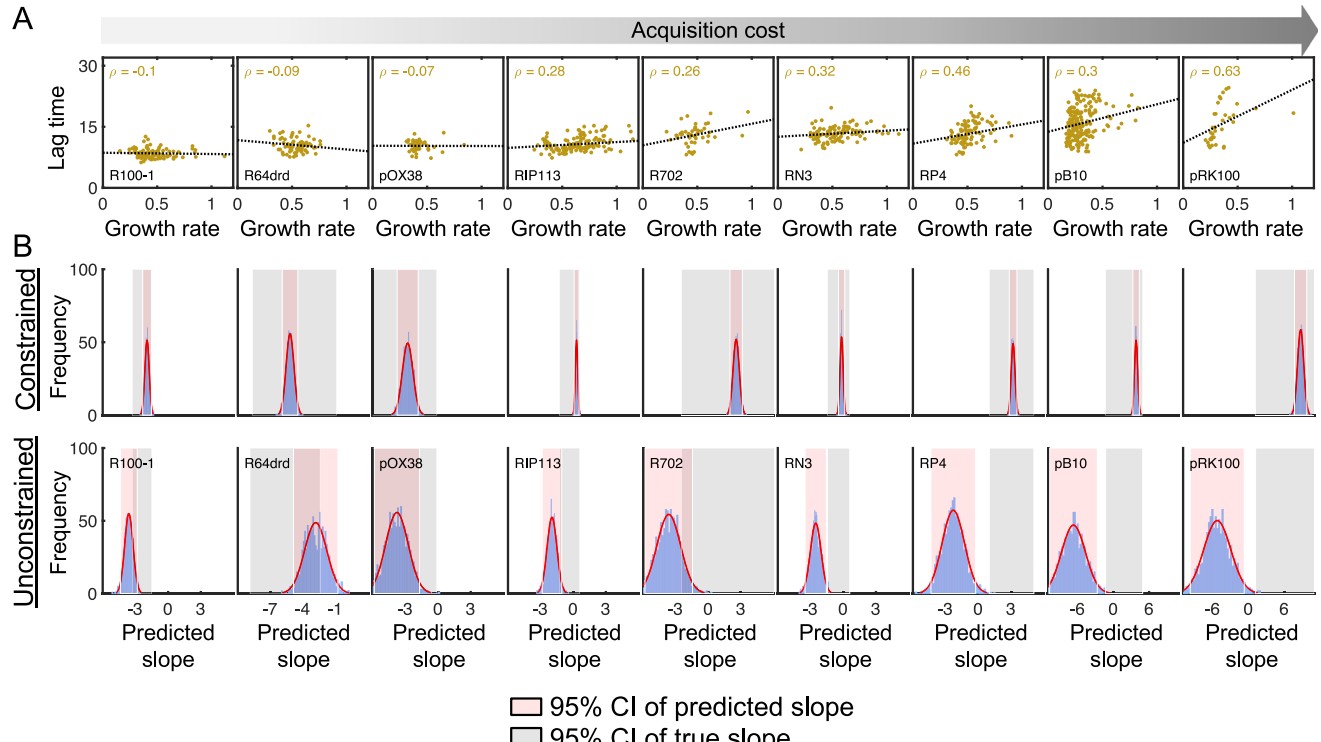

**Fig. 3 | Lag time and growth rate tradeoff as a function of acquisition cost. A** Lag time in hours (y-axis) as a function of growth rate (x-axis) for nine plasmids under tetracycline selection (i.e., Fig. 2A) shown in order of increasing acquisition cost (gray arrow, left to right). Individual markers represent de novo (yellow) colonies. Black lines represent the linear regression line of best fit, and the Pearson correlation coefficient ($\rho$) is reported for each population. Colony numbers are the same as those reported in Fig. 2A, and are listed in Supplementary Table 1c. **B** 300 clones with bootstrapped lag times were generated in 1000 independent rounds; in each round, associated growth rates that were either unconstrained (bottom) or constrained (top) by these lag times were also chosen. Parameters were used to simulate density over time using the modified Logistic equation, and the relationship between growth rate and predicted TTT was collected (see Supplementary Methods). X-axis is the TTT/growth rate relationship (i.e., the slope of the black line in the middle panel in Fig. 2B). Gray shading indicates the 95% confidence interval of the true slope for that corresponding plasmid, and red shading indicates the 95% confidence interval for the 1000 iterations. All Source data are provided as a Source Data file.

we assume that residual parents contribute to the total cell density of the initial population, but do not grow, consistent with our data (Supplementary Fig. 6b; Supplementary Text).

To characterize intrapopulation dynamics, we focused on changes in the overall population growth rate as a function of increasing initial cell density. Intuitively, higher initial cell densities (corresponding to additional parameter distribution sampling) simulates increasing clonal heterogeneity, and thus results in a greater chance of probabilistically selecting clones with longer lag times. In the absence of a lag/growth tradeoff (e.g., adapted transconjugants, or de novo transconjugants with no acquisition cost), these longer lag times are *not* correlated with higher growth rates, which remain equivalently distributed regardless of lag times. Thus, the population-level growth rate should remain constant at low to intermediate initial cell densities. At sufficiently high densities, individual cells will never reach their potential maximum growth rates due to the collective inhibitory effects of environmental carrying capacity; therefore, the observed population-level growth rate will monotonically decrease with increasing initial density. Consistent with this logic, modeling predicted and experiments validated that the growth rates of non-costly R100-1 de novo and adapted populations monotonically decrease with increasing cell density (Fig. 4B).

In the presence of a lag/growth tradeoff, however, as initial densities increase, higher lag times are, by definition, coupled with (i.e., constrained by) higher growth rates. These clones are disproportionately able to dominate a population, and thus, at low to intermediate initial cell densities, the population-level growth rate should increase with initial density. As in the no tradeoff case, at

excessively high initial cell densities, growth is constrained by carrying capacity effects. Together, then, for a sufficiently costly plasmid, we expected de novo cells to exhibit a biphasic relationship where growth rates increase at low to intermediate initial cell densities before decreasing at high initial cell densities. To verify this, we simulated and measured lag times/growth rates for the costly plasmid RP4 across identical initial densities as R100-1. Indeed, modeling and experiments confirmed a biphasic growth dependence in costly de novo, but not adapted, populations (Fig. 4C). Overall, these results demonstrate that for costly plasmids, a tradeoff between lag time and growth rate may enable heterogeneous populations to attain higher growth rates due to clonal selection. From an evolutionary perspective, then, this tradeoff may serve as a selective mechanism that favors variants/subpopulations optimal for HGT under a given set of environmental conditions.

Finally, we examined the ecological dynamics at the interpopulation level; that is, how does the lag/growth tradeoff impact selection outcomes of multiple genetically distinct competing plasmids, each with their own underlying phenotypic distributions? To answer this, we considered the scenario where a fixed, intermediate-cost plasmid population (Fig. 4D, red line) was competed against a second plasmid-carrying population exhibiting a range of acquisition costs. We then calculated the final percentage of the intermediate-cost plasmid population, both in the absence and presence of a lag/growth tradeoff. Simulations revealed that, on average, without any lag/growth tradeoff, a plasmid with a lower acquisition cost will always outcompete one with a higher acquisition cost (Fig. 4D, left). This finding is intuitive: lower-cost plasmids (i.e., shorter lag times) can dominate mixed populations earlier compared to costlier plasmids

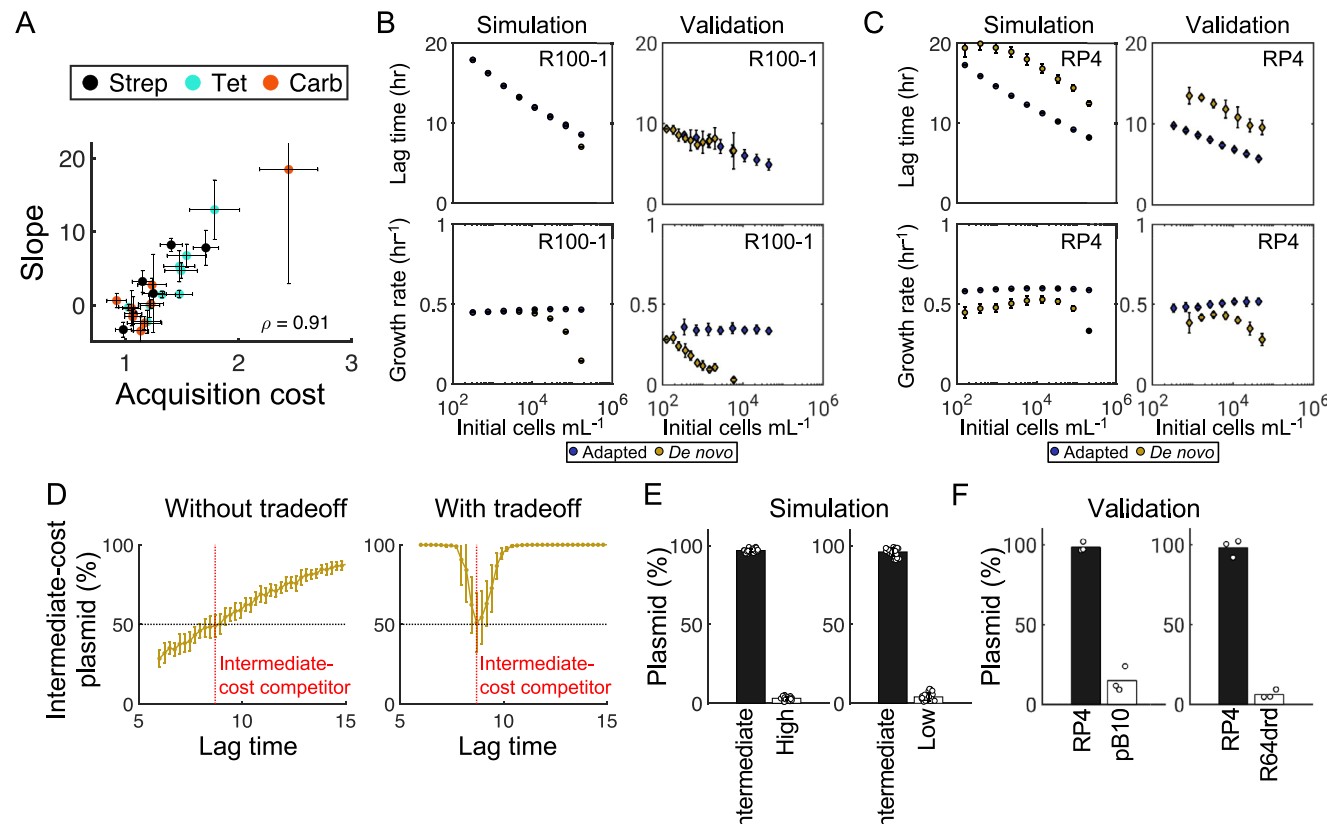

**Fig. 4 | Ecological/evolutionary advantage to intermediate acquisition costs.**
**A** Acquisition costs for all plasmids (i.e., Fig. 2D) with RB933 recipient is plotted against the slope of the regression line between growth rate and lag time (i.e., from Fig. 3A). The Pearson correlation coefficient is reported ($\rho = 0.91$). Orange, black, and blue marker colors correspond to plasmid selection with carbenicillin, streptomycin, or tetracycline, respectively. Markers represent the mean, and horizontal error bars represent the standard deviation, of acquisition costs for the indicated $n$ number of de novo colonies listed in Supplementary Table 1c; vertical error bars represent the 95% confidence interval corresponding to the slope estimates from Fig. 3A. **B** Left: Simulation predictions for the no-cost plasmid R100-1. X-axis is initial number of cells, and y-axis is the average population lag time (top) or growth rate (bottom). Right: Experimentally measured lag times and growth rates of R100-1 at corresponding initial densities (x-axis) for de novo (yellow) and adapted (purple) transconjugants. **C** Same as 4B for RP4. **D** Simulation predictions for pairwise competition outcomes between an intermediate-cost plasmid (red vertical line)

and one with varying acquisition costs, defined based on its lag time (x-axis), either in the absence (left) or presence (right) of a lag/growth tradeoff. Y-axis is the competition outcome, i.e., the percent of the intermediate-cost plasmid in the final population. **E** Simulation predictions between an intermediate-cost plasmid (black bars) and either a high-cost (left) or low-cost (right) plasmid (white bars). **F** Experimental competition outcomes between intermediate-cost plasmid RP4 (black bars) and either high-cost plasmid pB10 (left) or low-cost plasmid R64drd (right) (white bars). Plasmid percentage is calculated as the CFU from individual antibiotic-containing agar plates divided by the total CFU from tetracycline plates, multiplied by 100. For all simulations, marker values/bar heights represent means and error bars represent standard deviations, of $n = 50$ iterations; for experimental data in 4B, C, F, marker values/bar heights represent means and error bars represent standard deviations, of $n = 3$ biological replicates. For 4E–F, white markers indicate individual data points. All competition experiments use recipient strain BW25113-rif. All Source data are provided as a Source Data file.

(i.e., longer lag times), when the latter has no growth advantage. In contrast, when a lag/growth tradeoff is present (i.e., a longer lag time results in a faster growth rate for individual clones), simulations predicted that intermediate-cost plasmids will outcompete those with both low and high acquisition costs (Fig. 4D, right). This counterintuitive outcome is driven by optimality between lag time and growth rate: an intermediate-cost plasmid can outcompete both a low-cost and a high-cost plasmid due to the former's comparatively inferior growth rate, and the latter's excessive lag time. As such, we predicted that intermediate-cost plasmids would outcompete both their low- and high-cost counterparts.

To test this prediction, we used R64drd, RP4, and pB10 as representative low-, intermediate- and high-cost plasmids, respectively. These plasmids were chosen due to their distinct acquisition costs, (Fig. 2A and Supplementary Fig. 7), similar adapted growth rates/fitness costs (Supplementary Fig. 8a, b), and unique antibiotic resistance markers (Supplementary Table 1). First, we competed the adapted counterparts of each plasmid pair (i.e., R64drd-RP4 and pB10-RP4) to calibrate initial starting densities that would result in an equal

final ratio of the two plasmids (Supplementary Fig. 8c–e); doing so accounted for any differences in fitness effects and basal conjugation rates, thereby ensuring that any deviation from this equal ratio is driven entirely by de novo dynamics. Using the calibrated initial conditions, simulations predicted that in all cases, de novo RP4 (intermediate-cost) would outcompete both de novo R64drd and de novo pB10 (Fig. 4E). Indeed, experiments confirmed that the intermediate-cost plasmid RP4 dominated the population under all conditions tested (Fig. 4F). Overall, our results highlight that in mixed environments, due to the complex interplay between lag time and growth rate, intermediate-cost plasmids may be evolutionarily favored compared to their lower and higher cost counterparts.

## Discussion

Here, we used automated imaging to visualize single colonies of de novo transconjugants immediately following conjugation-mediated plasmid acquisition. Doing so allowed us to infer population-level trends and confirm our previous results[4]. Further, this approach allowed us to characterize the growth effects during the critical period

of plasmid establishment more quantitatively and with higher resolution than previous methods[4]. However, single-colony measurements also revealed significantly richer insights into the growth dynamics and underlying heterogeneity demonstrated by individual clones within a de novo population.

Our previous population-level measurements suggested that the acquisition cost manifests by either prolonging lag time and/or reducing growth rate. However, analysis at the single-colony level revealed that these observed effects likely arise due to underlying population clonal heterogeneity, whereby individual clones begin to grow at a wide distribution of times. Instead, the acquisition cost is primarily driven by lag time, and is largely independent of growth rate. This finding was general to 58 diverse conditions, encompassing various plasmids, antibiotics, and bacterial strains/species. Interestingly, recent works have found heterogeneous expression of resistance enzymes following conjugation, e.g., TetA in the presence of tetracycline[50,51], which may impact acquisition dynamics. Our genetic analyses did not suggest any obvious explanations for our observed tetracycline-specific differences across plasmids (e.g., both low and high costs); this, combined with consistent trends across multiple antibiotics, suggests our results are not driven by antibiotic-specific factors. However, drug-specific effects cannot be ruled out entirely; examining lag/growth dynamics in the absence of antibiotic selection altogether is an important next step.

This main finding—that lag time drives plasmid acquisition dynamics—is consistent with current understanding of lag dynamics. Specifically, during the lag phase, bacteria adapt to the selective pressures in their environment prior to exponential division[36,49,52,53]. This period also allows bacteria to withstand high stress, such as extended lag phases that enable bacteria to survive in the presence of antibiotics[54]. Thus, plasmid acquisition can be considered a natural extension of typical lag dynamics, which can be advantageous – here, reaching the steady state levels (e.g., copy number and gene expression) of the newly acquired plasmid *before* expansion allows the cells to maximize later growth.

Although lag time appears to drive overall acquisition dynamics, at the single-colony level, we found that clonal heterogeneity contributed significantly to population structure and ecological outcomes. Specifically, the growth rate was uncorrelated with the average plasmid acquisition cost at the population-level. However, within a single de novo colony, lag time exhibited a counterintuitive tradeoff with growth rate; that is, from the same population, individual colonies with longer lag times demonstrated higher post-lag growth rates compared to those with shorter lag times. This tradeoff suggests optimality between a plasmid's acquisition cost, lag time, and growth rate. Indeed, we believe the combination of all these factors ultimately drives new transconjugant selection, and dictates competitive plasmid outcomes. Our work showed one example of this - the selection of intermediate-cost plasmids compared to both their higher and lower-cost counterparts.

That plasmid costliness may provide bacteria with a growth benefit suggests that, unlike fitness costs, acquisition costs may never be fully selected against. Instead, maintaining some degree of costliness is beneficial for the population as a whole: it maximizes the diversity of competitors (plasmid-carrying and otherwise) over which a given strain may enjoy a competitive advantage. If so, this finding reveals a potential Achilles heel to exploit when determining novel strategies to modulate plasmid acquisition dynamics. For example, potentiating existing acquisition costs through genetic or metabolic manipulation may enable the targeted selection of specific plasmids of interest over others.

Overall, our results demonstrate that the acquisition cost can shape plasmid/strain competition and selection. Further, our work introduces greater complexity to the current understanding of plasmid costs, in that their presence may not always be disadvantageous. We note that our experiments used cells initiated from the stationary phase and simplified pairwise competitions. Future studies should expand the range of environmental conditions, physiological states, and population diversity, to assess how lag/growth outcomes may manifest in increasingly complex metabolic settings. Further, our preliminary analysis of plasmid genetics did not reveal any obvious molecular explanations for acquisition costs; a better understanding of the underlying mechanisms would allow us to integrate experimental measurements with more bioinformatic-driven methods to investigate acquisition costs across relevant environments and clinical settings. This type of integrative approach can help identify potential therapeutic pathways and provide additional insights into epidemiological trends of plasmid dissemination, furthering the development of novel plasmid control strategies.

## Methods

### Strains, media, and growth conditions

In all cases, experiments were initiated from single colonies picked from agar plates (BD Difco Luria-Bertani Agar Powder, cat #DF0445076), inoculated in 2 mL Luria-Bertani (LB) media (BD Difco Luria-Bertani Broth, cat #DF0446075), and incubated overnight at 37 °C for exactly 16 h shaking at 250 rpm. Where applicable, LB media was supplemented with specific antibiotics for selection: 15 µg/mL for tetracycline (Fisher Bioreagents, cat #BP912-100), 50 µg/mL for rifampicin (Alfa Aesar, cat #J60836), 50 µg/mL for streptomycin (Sigma-Aldrich, cat #S6501-50G), 50 µg/mL for kanamycin (Acros Organics, cat #61129-0050), and 100 µg/mL for carbenicillin

(Fisher Bioreagents, cat #BP2648-5); Supplementary Table 1 and Supplementary Table 3. In all cases, a colony-free LB negative control was included, and any experiment was discarded if growth was observed. All experiments were performed in M9 medium (M9CA medium broth powder from Amresco, cat # J864-100G, containing 2 mg/mL casamino acid, supplemented with 2 mM MgSO$_4$, 0.1 mM CaCl$_2$, and 0.4% w/v glucose). A list of all strains and plasmids used in this study, along with all main experiments and corresponding antibiotics used for selection, can be found in Supplementary Tables 1 and 3.

### Susceptibility screening for plasmid resistance

Comprehensive resistance profiles were determined for all plasmids used. First, each plasmid was transferred by conjugation to the spontaneous rifampicin-resistant mutant of BW25113, denoted BW25113-Rif. As this strain is only resistant to rifampicin, full susceptibility profiles of plasmids could be obtained. For each plasmid, 1 mL LB cultures containing the appropriate selection drug were prepared. Cultures were inoculated, and cells were grown in a 96-well plate as described above. After overnight incubation, all cultures were diluted 1:1000 in M9CA media. About 5 µL of each diluted plasmid was spotted on 12 square agar plates. Eleven of the twelve plates contained exactly 25 mL of agar supplemented with the appropriate antibiotic (Supplementary Table 1): amoxicillin/clavulanate (60 µg/mL), carbenicillin (50 µg/mL), chloramphenicol (30 µg/mL), ciprofloxacin (1 µg/mL), gentamicin (50 µg/mL), kanamycin (50 µg/mL), rifampicin (50 µg/mL), spectinomycin (50 µg/mL), streptomycin (50 µg/mL), tetracycline (15 µg/mL), or trimethoprim (10 µg/mL). The 12th plate served as a drug-free control. About 5 µL of a negative M9CA control was also spotted on all plates to assess for contamination. Plates were grown for 24 h at 37 °C, and susceptibility was recorded. Following an initial screening, resistance profiles were regularly checked using 2 mL LB overnight cultures to confirm plasmid identities. Recipient BW25113-Rif was also regularly screened to confirm rifampicin resistance, but sensitivity to all other drugs.

### Generating de novo and adapted transconjugants

Individual donor and recipient clones (either RB933 or BW25113-rif) were grown overnight, as described above. After 16 h, cultures

were spun down at 10,000 rpm for 1 min, resuspended 1:1 in M9CA, and incubated at room temperature for 5 min. Equal volumes of donor and recipient were then combined, and donor-recipient mixtures were incubated for 1 h at 25 °C to allow for conjugation. Following this conjugation period, the mixture was immediately vortexed for at least 5 s to disrupt the mating bridge, at which time the mixture was considered to contain de novo transconjugants for subsequent experiments. To generate adapted transconjugants, the conjugation mixture was then serially diluted in a 96-well plate; 10 µL was spotted onto agar plates containing appropriate dual antibiotics, and grown overnight for 16 h at 37 °C. Dilutions that achieved single-colony growth were recorded and used to calibrate plating densities for subsequent acquisition cost experiments. Colonies on these plates were considered adapted transconjugants thereafter, and were maintained either on agar plates for immediate use, or stored in 25% glycerol for subsequent streaking and experiments.

### Quantifying the plasmid acquisition cost

De novo transconjugants were generated as described above. In parallel, a clone of an adapted transconjugant was also grown from a streaked or spotted agar plate and maintained under identical conditions (i.e., resuspended in M9CA and incubated at 25 °C for 1 h). After the 1 h incubation, both de novo and adapted transconjugants were simultaneously diluted. For each population, three dilutions were spotted on two sides of a single agar plate containing appropriate dual antibiotics in 3-4 technical replicates. For de novo transconjugants, plasmid-specific calibration experiments as described above were used to determine appropriate dilutions that would span one order of magnitude above and below that which 3–30 colonies were expected to grow (Supplementary Table 1); in this way, any day-day variation was reliably captured. Adapted transconjugants were always plated in dilutions of $10^5$–$10^7$. After spotting, plates were immediately dried for 5 min at 30 °C, and then placed face-down onto Raspberry pi-automated EPSON v800 flatbed scanners housed in a 37 °C incubator. Automated images were taken every 15 min for at least 24 h using the linux "scanimage" command, and analyzed using a custom MATLAB pipeline. All acquisition cost experiments were performed in at least biological duplicates, and the majority in triplicate (see Supplementary Table 1c for exact numbers). For each plasmid, the acquisition cost was quantified by pooling all colonies across repeated experiments, and normalizing the TTT for each de novo colony by the average TTT of all adapted colonies; statistical significance was determined based on $p$ values resulting from adapted compared to de novo TTT using a two-sided t-test with Bonferroni correction when relevant. The magnitude of the acquisition cost for each plasmid was the average of this normalized value.

### Image analysis

The custom image analysis pipeline utilized throughout this study consists of the following steps: first, x-y coordinates and radii are collected for every spot corresponding to a 10 µL drop containing countable colonies into a master spreadsheet, along with relevant metadata (plasmid type, antibiotic, etc.); this allowed us to match each colony with its corresponding spot for subsequent verification of density effects (i.e., number of colonies per spot). Next, we perform several validation steps to ensure each detected colony is valid. First, we use the final time point of each experiment to generate a binary mask based on global thresholding. Each object that is detected in this mask is classified as a 'colony' or 'not a colony' using a simple binary bag-of-features classifier. This classifier was trained on ~5000 images of colonies and non-colonies, where positive colonies were primarily high-quality, centered, round objects, and

negative colonies were various plate artifacts such as writing, scanner jitter, condensation, etc. Following this initial classification, isolated colonies are further refined using manual input to correct obvious misclassifications, i.e., to manually accept "valid" colonies or reject "invalid" colonies. For example, a small number of colonies that do not appear perfectly round may be misclassified as "invalid" and would be inappropriately rejected by the classifier. We note this manual input is done on a colony-by-colony basis based purely on appearance, and is therefore blinded to whether the colony is adapted or de novo. Once this process has determined the set of all valid colonies, we iterate over the entire time series for each colony position. At each time point, thresholding is used to get a binary mask; the resulting image is passed through another bag-of-features classifier that is trained to classify images as either "noise" or "not noise", trained on a library of approximately 20,000 images. If an image is classified as "noise", it will not be analyzed further; if it is classified as "not noise," the size of the colony is determined according to the number of binary pixels. This is done for all colonies and all time points, and forms our raw growth curves. These raw growth curves, along with associated metadata, are stored in MATLAB structures for post-image processing. During post-processing, growth curves are further filtered for obvious outliers by removing any curves with a final density (i.e., pixel number) outside of two standard deviations of the average for that colony type (i.e., only within de novo and adapted from a single plate) and experiment; this is a conservative estimate, since these often corresponded to colonies that appeared round in a mask but consisted of >1 colony in extremely close proximity that passed through the classification step), or those that did not change in size significantly over the duration of the experiment (i.e., something that appeared circular but was a shadow and not a true colony). Finally, curves are smoothed by connecting spikes due to shadowing or pixelation with averages of the pixel values at the time points immediately before and after. Curves with spikes that cannot be reliably closed (greater than two time points) are removed to avoid unintentionally distorting observed lag times. Additionally, curves that did not reach a sufficient plateau (based on its derivative) are also removed to avoid inaccurate growth rate estimates from logistic fitting. Combined, these filtered datasets were stored and used for all subsequent quantification.

### Growth curve quantification

Image-derived growth curves from all filtered datasets, and all plate reader growth curves, were fit using the modified Logistic growth equation:

$$N = \frac{A}{1 + \left(e^{\frac{4\mu_m}{A}(\lambda-t)+2}\right)} \tag{1}$$

where $\mu_m$ and $\lambda$ are taken to be the maximum growth rate and lag time, respectively, $N$ is the log-transformed cell density, and A is the maximum density achieved. Lag times calculated in this way were highly correlated with observed lag times as the first time point where a pixel intensity was greater than 0. Finally, the time-to-threshold (TTT) was determined by obtaining the closest time the pixels reached the defined threshold of 0.8. We note that the choice of threshold did not impact any results (Supplementary Fig. 1b, 2a). Finally, acquisition costs are quantified by dividing the time-to-threshold of de novo transconjugants by the average time-to-threshold of adapted transconjugants. Manual growth rates (Supplementary Fig. 3a, b) were obtained using curve-smoothing and numerical differentiation to identify the time $t$, at which the maximum derivative occurred within the region of exponential growth. The maximal growth rate was then calculated as the linear slope of the tangent line passing between the two time points above

and below time $t$, i.e.:

$$\mu_m = \frac{(t+2)-(t-2)}{y_2 - y_1} \qquad (2)$$

where $y_2 > y_1$. To quantify the instantaneous growth rate (Supplementary Fig. 3c), colony size was determined by summing the pixel number per colony and averaging the size across all colony replicates within a single-plasmid population. The instantaneous growth rate (hours$^{-1}$) was calculated as $\log(x(t)/x(t\text{-}1))/0.25$, where x is the number of pixels at time t. Instantaneous growth rate reports the change in each colony size at every time point as a function of the overall average colony size, which provides an instantaneous estimate of colony expansion.

## Comparing tetracycline-specific genetic features with acquisition costs

Plasmids were isolated using the ZymoPURE II Midiprep Kit (cat #D4200) according to the manufacturer's instructions; sequences and de novo assemblies were generated using commercial vendors (either the MGH DNA Core or plasmidsaurus), and verified using an in-house pipeline. When applicable, provided assemblies were compared with reference sequences to verify sequence identity. The following plasmids were exact matches to their references: RP4 (BN000925.1), pRK100 (CP060383.1), and pB10 (AJ564903.1). The remaining plasmids can be found under the following GenBank accession IDs: R100-1 (OQ683449), R64drd (OQ683450), R702 (OQ683451), RIP113 (OQ683452), RN3 (OQ683453), and pOX38 (OQ683454). Tetracycline resistance proteins were identified using NCBI BLASTp by querying the plasmid's translated protein sequences against a database of tetracycline resistance (TetA, class B) and tetracycline repressor (TetR, class D) *E. coli* protein sequences. The tetracycline operon nucleotide sequence was extracted from each plasmid's annotated GenBank file, from the beginning of the first tetracycline resistance gene to the end of the last one, which captured the operator sequence between *tetR* and *tetA*. Then, CLUSTALW 2.1 was used to generate a multiple sequence alignment of the tetracycline operon for each plasmid[55]. Alignments and distance matrices were visualized in Genieous Prime (Supplementary Fig. 2c and Supplementary Table 2).

## Intrapopulation competition experiments

To assess the effect of clonal heterogeneity amongst a single-plasmid population, the no-cost plasmid R100-1 and intermediate-cost plasmid RP4 were used. Experiments for adapted and de novo transconjugants were performed identically as in all acquisition costs, with the following modifications. Following the 1 h incubation at 25 °C, instead of plating onto solid agar, cells were diluted into M9CA supplemented with tetracycline/rifampicin. Dilutions were determined to allow for de novo and adapted densities of both plasmids to equal another, with initial cell numbers ranging from 100 to 1,000,000 cells. These calculations were obtained from colony-forming unit (CFU) measurements averaged across at least three biological replicates. In all cases, the average CFU of adapted populations was statistically insignificantly different from 2E + 09; this number was therefore used to estimate dilutions for all adapted plasmid conditions. 200 μL of diluted cells were aliquoted into wells of a microtiter plate and covered with 50 μL of mineral oil. The plate was placed in a temperature-controlled 37 °C Tecan plate reader and read kinetically every 15 min for 24 h. Data were collected on Tecan's open-source iControl software (V3.6.01). To verify initial densities during the experiment, the CFU of each plasmid was also taken at the same time on a tetracycline/rifampicin agar plate. For these experiments, RB933 was used as the host strain. All experiments were conducted in technical quadruplets for three biological replicates.

## Quantifying plasmid fitness costs

To compare growth rates of adapted transconjugants, overnight cultures of individual clones of adapted RP4, pB10, and R64drd in strain BW25113-rif were diluted 1:1000 in M9CA supplemented with tetracycline/rifampicin. Diluted cells were added to a 96-well plate in technical triplicates or quadruplets. Each well was covered with 50 μL of mineral oil, and the plate was read kinetically in a temperature-controlled (37 °C) Tecan plate reader for 24 h (Supplementary Fig. 8). All experiments were conducted with at least three biological replicates. Growth rates were determined using the modified Logistic model described above.

## Determining the impact of parent populations on the carrying capacity

To quantify the effect of background cells (e.g., residual donor and recipient parents) on the transconjugant carrying capacity, overnight cultures of donor pB10, recipient BW25113-rif, and the adapted transconjugant were grown as previously described. The donor and recipient strains were conjugated as described above. Following incubation, conjugation mixtures containing de novo, donor, and recipient cells, were diluted threefold into the top row of a 96-well plate pre-filled with M9CA media containing both tetracycline and rifampicin; cells were then serially diluted threefold such that the estimated number of de novo cells reached a final dilution of ~6560 cells/well. In parallel, individual donors and recipients were diluted at the same density to confirm antibiotic sensitivity. Every well was then covered with 50 μL mineral oil, and the plate was placed in a Tecan plate reader maintained at 37 °C; kinetic measurements were taken every 15 min for 24 h (Supplementary Fig. 6b). All populations were plated in technical triplicates. Growth rates were fit as previously described using the modified Logistic model.

## Interpopulation competition experiments

For competition experiments, plasmids were competed using the host strain BW25113-rif. Recipient strain RB933 was not used for competition experiments, since this strain has multiple chromosomal resistance markers, which would have prevented distinguishing plasmids via antibiotic selection. Acquisition cost trends were verified to be consistent for each plasmid across recipient strains (Supplementary Fig. 7). Plasmid pairs for competitions were chosen to test the prediction that an intermediate-cost plasmid may dominate both a low- and high-cost plasmid. To do so, we identified three compatible plasmids that had distinct acquisition costs and met the following criteria: (1) adapted transconjugants were to have similar growth rates/fitness costs, (2) plasmids were to have at least one unique resistance marker each, and (3) plasmids were to share resistance to tetracycline. R64drd against RP4 for low-intermediate, and pB10 against RP4 for high-intermediate, met these criteria (Supplementary Fig. 8a, b). Competitions of adapted transconjugants were first conducted to determine the appropriate initial mixing density for de novo experiments; this allowed us to account for any growth- or conjugation-specific effects. Briefly, overnight cultures of appropriate adapted transconjugant plasmid pairs were incubated separately for 1 h and at 25 °C as described previously. RP4 and either pB10 or R64drd plasmids were then diluted at various densities, and combined in a 96-well plate. The plate was sealed with sealing paper (Diversified Biotech, cat #BEM-1), and incubated for 24 h in a 37 °C incubator with agitation at 250 rpm. At time = 0, the CFU of each individual plasmid population was measured using tetracycline/rifampicin agar plates, alongside unique resistance marker plates (carbenicillin/rifampicin for RP4, streptomycin/rifampicin for both pB10 and R64drd) to verify initial mixing densities. At the end of 24 h, competition mixtures were serially diluted and plated onto the same three unique drug plates such that single colonies were visible. All plates were grown for 16 h at 37 °C. After determining that an initial 1:1 mixture of RP4 to pB10, and a 1:20

mixture of RP4 to R64drd, yielded an equal final ratio after the competition (Supplementary Fig. 8c, d), de novo competitions were performed using the same initial mixing ratios. In brief, de novo transconjugants of RP4, pB10, and R64drd were prepared, and combined at their respective adapted calibrated ratios. The CFU was taken of each plasmid's transconjugant at 0 and 24 h, using the three unique drug plates. Plasmid percentages were calculated by taking the CFU from each unique drug plate (i.e., carbenicillin/rifampicin for RP4 or streptomycin/rifampicin for R64drd and pB10) divided by the total CFU from tetracycline/rifampicin, and multiplied by 100. Each competition was conducted with at least three biological replicates.

### Reporting summary

Further information on research design is available in the Nature Portfolio Reporting Summary linked to this article.

## Data availability

All data generated in this study are provided in the Supplementary Information and/or as a Source Data file. The plasmid assemblies generated in this study are available under the following GenBank accession IDs: R100-1 (OQ683449), R64drd (OQ683450), R702 (OQ683451), RIP113 (OQ683452), RN3 (OQ683453), and pOX38 (OQ683454), and archived on the Zenodo database at the following DOI: 10.5281/zenodo.7753403. Source data are provided with this paper.

## Code availability

All tetracycline analysis code is archived on Zenodo at the following DOI: 10.5281/zenodo.7753403 (https://zenodo.org/badge/latestdoi/606017397). It is also available on the lab's Github: https://github.com/ajlopatkin/acquisition_cost_tet_analysis. All image analysis code is also archived on Zenodo at the following https://doi.org/10.5281/zenodo.7753412 (https://zenodo.org/badge/latestdoi/557526918) and on the lab's Github: https://github.com/ajlopatkin/acquisition_cost_image_analysis.

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

## Acknowledgements

We thank I. Gordo, D. Mazel, F. Dionisio, J. Alves Gama, E. Top, and M. Bruckner for their generous gifts (See Supplementary Methods). We also thank Maya Fabozzi from the Lopatkin Lab for generating the rifampicin-resistant mutant of the BW25113 strain. This study was funded by the National Science Foundation award #2040697 (A.J.L.) and the National Institute of Health awards #1R15GM143694-01 (A.J.L.) and #R01AI150152 (B.T.).

## Author contributions

M.A. conceived the research, designed and performed all experiments, conducted data analysis, and wrote the manuscript. H.P., J.B., and S.E. designed and performed experiments and conducted data analysis. A.G.-S. assisted with modeling and manuscript editing. A.G.-S. and A.-C.U. provided the KPN strain and assisted in manuscript editing. B.T. assisted with data interpretation and manuscript editing. A.J.L. conceived the research, designed experiments, performed all modeling, interpreted the data, and wrote the manuscript.

## Competing interests
The authors declare no competing interests.
