## [Peer Review File · Nature Communications]

Reviewers' Comments:

Reviewer #1:

Remarks to the Author:

The authors explore the effects and consequences of de novo plasmid acquisition costs by tracking single colonies across a wide range of conditions. They find that there is a trade-off between adaptation time to the plasmid and cell growth, which can have important consequences for the dynamics of plasmid spread. The study addresses a timely and important question on the dynamics of plasmid spread and is overall comprehensive and well written. However, some parts, especially figures and legends, are lacking in clarity or not to be corrected and as several of the formulas were not displayed correctly, and it was hard to judge if they were appropriately chosen.

Major points / questions:

L274 – 299: This part is hard to understand, particularly as it does not match with the referenced figures (see below). It is unclear to me if the goal is to investigate the success of the plasmid or the overall (heterogeneous) population (L297) as it is not explained properly what the heterogeneous population is. If I understand it correctly, then the 'background' population is non-growing (i.e. inhibited by antibiotics), which doesn't seem to reflect competition and collective inhibitory effects very well?

Figure 4 B&C: This part of the figure does not seem to fit with the description in the main text and the legends are not detailed enough to understand what the figure parts are showing, specifically because there are different symbols used and it is not clear which ones are the simulations (I would have guessed Prediction, but then there are also symbol with error bars in B but not in C). Further, B and C seem to be switched, at least according to the main text (B should show R100-1 and C RP4).

4D: Again, the legend is not enough to understand what is shown and while the text states that intermediate acquisition costs (and therefore intermediate lag time) are favored, the right plot seems to show that intermediate lag times have the lowest percentage of plasmids?

4E&F: It is confusing that high and low panels are switched in E and F (i.e., high-low in E and low-high in F).

SI Table1 gives a good overview over the plasmids used, including their main characteristics, but no analysis was performed to prove or disprove any correlation between those traits (e.g. length or conjugation rate) with acquisition cost or growth rate.

If acquisition costs mainly affects lag time, how important is that effect in an exponentially growing culture, i.e. when bacterial cells are already metabolically active, or is lag time here more thought of as the time it takes for an individual cell to start growing after it received the plasmid?

The methods section is generally very detailed and well written, but several of the formulas in the main text and the SI are not displayed correctly (only empty squares), so they were not readable and impossible to review.

Minor comments:

The second part of the introduction could be a bit more focused. It is relatively dense and mentions many concepts from molecular processes of cell metabolism to selection in clonally heterogeneous populations, which makes it a bit hard to take away which gaps and questions are addressed (as for example biomolecular processes as well as complex microbial communities are not really touched upon for the rest of the paper). Similarly, the importance of distinguishing the lag time and growth rate parts of the acquisition cost does not come out clearly.

L51: 'facilitate' seems to be a weird word choice here

L85: 'heterogenous' populations are mentioned at several points throughout the manuscript but it is not quite clear what they refer to and why they are important

L181ff: It is unclear what instantaneous colony growth is (compared to the otherwise reported growth rates) and how it relates to 'equivalent growth rates'.

Also, in Fig. S3C it looks like instantaneous growth rate is decreasing with acquisition cost for de novo populations – how does that fit with the trade-off between lag time and growth rate?

L192f: For RIP113 the correlation between TTT and growth rate seems to change from neutral / decoupled to positive but maybe that is not significant?

L215: Detrimental for plasmid acquisition or establishment? The plasmid was still acquired, it might not be able to establish in the population though.

L225: Was only mean or the whole distribution used (as the variance in TTT seems to be quite high in costly de novo populations)?

L271: What do iterations correspond to here (in terms of generations)? Also, maybe give an intuition for why you are using (this) model as you do in all the other sections very nicely.

L 347: Again, it is not entirely clear what is meant by heterogeneity (within a de novo plasmid population, or in a population containing de novo and adapted plasmids, or a population of recipients and donors?) and how it relates to the observed results.

L354ff: The conclusion that acquisition cost might be beneficial is confusing as a tradeoff suggests that it cannot be selected against without losing beneficial growth benefits. Hence, growth benefits would be selected for, but not higher acquisition costs per se.

Figure 1 C-E: TTT should be explained in the figure and/or the figure legend and time units should be given.

Figure 2: Red and green is easily the worst color choice here and as it is showing different things across the panels, those colors should not be the same. Generally, the colors in B and C do not really add additional information as they are too hard to distinguish (the only visible ones look similar to the colors used in D and E but they show different things, so that only adds confusion), so it might be better to just show them in black.

Titles for the individual panels would help

Figure S3A: growth rate on the y-axis is likely wrong

Figure S6B: More explanation would be good, I assume only the growth of the de novo transconjugants is shown and the rest of the population was suppressed by antibiotics?

S6C: There is no yellow in the figure but it is mentioned in the legend?

Reviewer #2:

Remarks to the Author:

In the manuscript "Tradeoff between lag time and growth rate drives plasmid acquisition costs", Ahmad and co-workers investigate the physiological effects associated with the acquisition of conjugative plasmids. This is an important topic as conjugation plasmids are major contributors to the dissemination of metabolic properties in bacteria, including drug resistance. Studying the effect of plasmid acquisition on the recipient strain is essential to understand the dynamics of bacterial populations undergoing conjugation and the parameters/rules of plasmid success. They use both experimental approach and mathematical modelling to compare, at the single colony level, the growth of "de novo transconjugants" (that have just acquired the plasmid) to the growth of adapted transconjugants (~24h after plasmid acquisition) (These two types of transconjugants have been defined and characterized in previous reports from the Lopatkin lab). This approach is much more powerful and informative than simply comparing plasmid-containing strains to the plasmids-free counterpart strains, and represents an obvious advantage compared to liquid culture analysis. Crucially, this work analyses growth in terms of lag time and growth rate, and a range of nine conjugative plasmids (from various incompatibility groups) are tested.

The results presented provide a comprehensive description of plasmid acquisition effects at the single-transconjugant colony level (generated by single-transconjugant cells) with unprecedented

amounts of details and in-depth analysis. The work demonstrates that for a given costly plasmid, transconjugant clones with longer lag time eventually exhibited a faster growth rate, suggesting the existence of a tradeoff between the adaptation period and the final growth ability. The findings are compelling (thanks to robust statistical analysis, essential controls regarding the antibiotic used for plasmid selection, validation by mathematical modelling, and the reproducibility through the plasmid range and *E. coli* or *K. p* stains), and the implication for plasmid ecology and dissemination success are well discussed. The data are well presented and the interpretations well supported by the results. However, the authors could consider discussing molecular mechanisms potentially involved in the observed heterogeneity between clones with the help of published single-cell studies.

Overall, this reviewer thinks this work is an original and important addition to the field of conjugation plasmid biology, both in terms of methodology and findings.

Comments:

1- There is no indication of overall conjugation efficiency for the different conjugative plasmids tested. Results from plating de novo transconjugants on dual-antibiotic agar plates should have given this information. The authors could consider providing conjugation efficiency estimates in the supplementary figure, if relevant and possible. This is obviously not the focus of the present work, but it still would be helpful information. This could also reveal any potential correlation between the transferability and acquisition cost of the tested plasmids. Have the authors investigated such a correlation?

4- L. 184. This is the first mention of "tetracycline selection". Tc selection was also probably used to generate RP4 results in Figure 1!? This should probably be stated earlier in the text. The rationale/implication of using Tc selection for the assays should also be discussed. Indeed, the experiment might or might not be possible with a bactericidal rather than a bacteriostatic antibiotic. The implication of using a translation-inhibitor (rather than a transcription-inhibitor, or DNA-damaging antibiotic, or cell-division inhibitor) might also deserve a comment.

5- In the assay used, de novo transconjugants cells are in contact with Tc at the time of plating on dual-selection only. It follows that the host's multidrug efflux pumps (in particular AcrAB-TolC) might also play a role during plasmid establishment in the presence of antibiotics that are a substrate for these chromosomally-encoded pumps. A such role for multidrug efflux pump during plasmid acquisition under selection pressure has been demonstrated previously (PMID: 31123134) and could be considered/discussed here. This would be even more relevant than the authors did an excellent job by validating their results using other antibiotics for plasmid selection (streptomycin or carbenicillin) and different bacterial strains (*Klebsiella pneumoniae*).

6- L. 211-221. Regarding the heterogeneity of behaviour among individual clones, which the authors summarize as "as lag time increases, so does growth rate". The previously mention report (PMID: 31123134) also reveals the heterogeneity in the level and rate of production of the plasmid-encoded TetA efflux pump that confers resistance to tetracycline after plasmid acquisition (F in this case). In a further report, these different TetA production levels in transconjugants (corresponding to adapted transconjugants in this work) were correlated to the Tc-resistance ability at the single-cell level (PMID: 32761242). Do the authors think that the observed growth (lag time and growth rate) heterogeneity of individual clones could be related to plasmid-genes expression heterogeneity? This could help discuss molecular mechanisms underlying the behaviour revealed in this report. Can the authors think of any other molecular mechanism that could account for or contribute to the observed phenotypes?

Minor points:

1- All reference numbers are after full stops throughout the text, "diversity and evolution.1,2 Conjugation facilitates...". It should probably be "diversity and evolution^{1,2}. Conjugation facilitates..."

2- In the introduction, the three following statements are justified but sound quite repetitive and rhetorically redundant. The authors might want to rephrase.

"Thus, it is critical to understand factors that facilitate plasmid dissemination to identify strategies to predict, prevent, and/or control such occurrences."

"Therefore, understanding the effects of plasmid acquisition is critical to accurately predict the overall potential and success of conjugative plasmids."

"Yet, this distinction is critical to better understand both the physiological manifestation of acquisition costs, ..."

3- (L. 120-121) "that single colonies are visible". At this stage, it would help to indicate "that single transconjugant colonies are visible".

4- (L. 141) "more rigorous than the previous methodology". The authors might want to change "rigorous" to another word that does not undermine the rigour of previous approaches, which were also rigorous though much less precise and resolute.

5- Log cell density plots (Fig. 1C, Fig. S1A...) or corresponding legends should indicate the number of individual colonies monitored and plotted.

6- L. 164-166. "Plasmids with no or low acquisition costs (e.g., R100-1, R64drd, and pOX38)". It would help the reader to also put between brackets the list of what is interpreted as "high-cost plasmids exhibited a decoupling or positive correlation between growth rate and TTT".

Reviewer #3:

Remarks to the Author:

Comments for Authors

The manuscript by Ahmad et al. describes a rather interesting (after reading this) and clearly understudied phenomena: the plasmid acquisition cost. Overall, the paper very clearly shows that there is a scientifically interesting "story" happening at the very moment when a bacterium receives a new genetic element. The correlation of increased lag time with the higher growth rate reveals something about the acute interaction between cells and their (sometimes) transient genetic hitchhikers. The ecological considerations of the manuscript add insights on the population and community level, and can help us understand why acquisition costs both vary and may be selected for in different ecological and environmental setups. The paper is well-written and clear. First figure is very informative and makes the paper much easier to approach. Data appears to be consistent and well-presented. Ultimately, of course, it would be interesting to know the interactions on molecular level that are responsible for the extended/decreased lag period and the subsequent growth rate. These, however, are very difficult to study. Nonetheless, this paper also contributes to the general theme on the effects that foreign genetic elements (even if relatively well adapted to that organism) cause in its new genetic environment. For example, it would be interesting to know how the acquisition costs would evolve in plasmids that are originally non-adapted to a particular host. Such changes could also reveal about the reasons behind the observed acquisition costs. Further, the manuscript suggests ways to decipher why certain plasmids are doing better than others in clinical or other relevant environment where plasmids may play a notable role. As such, clinical plasmids could be more often measured for their acquisition costs, and we could see whether there are correlations with the plasmid's success (prevalence). Altogether, this reviewer finds no flaws in the manuscript, and it turned out to be quite surprising and insightful. As such, this reviewer would recommend the manuscript for publication, given the corrections suggested below.

Minor changes:

Line 145: "...intrapopulation heterogeneity not previously observed..." This reviewer would rather say "not previously quantified", as most bacteriologists would know that very often any plated cells grow colonies at the same rate.

Methods: For the LB powder etc. just the catalogue numbers are provided without the manufacturer.

Growth curve quantification: first equation has some missing symbols.

Reviewer #1 (Remarks to the Author):

The authors explore the effects and consequences of *de novo* plasmid acquisition costs by tracking single colonies across a wide range of conditions. They find that there is a trade-off between adaptation time to the plasmid and cell growth, which can have important consequences for the dynamics of plasmid spread. The study addresses a timely and important question on the dynamics of plasmid spread and is overall comprehensive and well written. However, some parts, especially figures and legends, are lacking in clarity or not to be corrected and as several of the formulas were not displayed correctly, and it was hard to judge if they were appropriately chosen.

We thank the Reviewer for the overall positive evaluation of our work, and for their thorough read of the paper. We have addressed all the major and minor comments raised below. We hope the Reviewer agrees the updated manuscript is much improved and now suitable for publication.

Major points / questions:

1. L274 – 299: This part is hard to understand, particularly as it does not match with the referenced figures (see below). It is unclear to me if the goal is to investigate the success of the plasmid or the overall (heterogeneous) population (L297) as it is not explained properly what the heterogeneous population is. If I understand it correctly, then the 'background' population is non-growing (i.e. inhibited by antibiotics), which doesn't seem to reflect competition and collective inhibitory effects very well?

We apologize for the confusion in this section, particularly due to the use of the word heterogeneous. Briefly, flatbed scanning allowed us to observe the heterogeneity associated with growth characteristics of individual colonies, which we refer to in the text as "clonal heterogeneity"—that is, phenotypic differences amongst single colonies otherwise belonging to a genetically identical population (**Line#147-148**). We use the term clonal heterogeneity in this context, as it is consistent with other works focusing on similar questions in the literature¹⁻⁸. Particularly, we noticed that *de novo* transconjugant colonies belonging to the same population exhibited clonal heterogeneity in their lag times and growth rates. Moreover, for *populations* with higher *average* acquisition costs, individual clones with longer lag times also had higher growth rates. This tradeoff could be captured by a slope between the growth rates and lag times of all clones within a population (Figure 4A); in other words, the relationship arises *because* of the clonal heterogeneity present.

The goal then, in Figure 4B-F, was to assess how this clonal heterogeneity, and specifically the lag/growth tradeoff, would impact ecological outcomes of the population. To investigate this, we focused on two different levels of selection: (1) within a single population (i.e., a genetically identical population with one plasmid), which we refer to as intrapopulation (Figure 4B-C), and (2) across multiple populations (i.e., two mixed, genetically distinct populations, each with a unique plasmid), which we refer to as interpopulation (Figure 4D-F). In all cases, experiments and simulations were designed to examine the overall population-level behavior as driven by their underlying individual clonal makeups. In the case of intrapopulation experiments, we focused on characterizing features of the overall population itself (e.g., population-level growth rate and lag time), whereas in the case of interpopulation experiments, we focused on characterizing features of the competition (e.g., the plasmid outcome of pairwise competitions).

Based on the Reviewer comment, we realized there may be some misunderstanding in the text regarding the role of the background population. Indeed, to examine dynamics of *de novo*

transconjugants, in all cases, the conjugation mixture (consisting of donors, recipients, and freshly generated *de novo* cells) is diluted into dual-antibiotic media. Although this mixture contains residual donors and recipients, the use of two antibiotics inhibits the growth of the parents that make up the background (Figure S6B). Since cell death occurs on relatively shorter time periods than the dynamics of interest (e.g., plasmid competition/growth rate), and since we accounted for these background effects in our modeling, we assume the contribution of residual parents at the start of the experiment to be negligible. Thus, “competition” refers to that between transconjugant cells. In response to this and other comments below from this Reviewer, we have more clearly defined the use of ‘heterogeneous’ throughout the manuscript; one example is shown below and highlighted in blue throughout:

Line#146-151: “Aside from validating previous work, colony tracking also revealed extensive clonal heterogeneity within the *de novo* population; here, we use heterogeneity to denote individual colonies that belong to the same genetically identical population but exhibit phenotypic differences, as evidenced in the variability of growth curves for individual RP4 *de novo* transconjugant colonies. Because such clonal differences would not be detectable in liquid-culture, this heterogeneity had not been previously quantified.”

We modified the introduction to the ecological/modeling section to clarify the purpose/goal of the simulations:

Line#280-284: “Thus, we sought to assess how the lag/growth tradeoff, which is inherently dependent on underlying clonal heterogeneity, would impact selection dynamics and ecological outcomes. We first examined selection dynamics at the intrapopulation-level; that is, how does the lag/growth tradeoff impact clonal selection within a genetically-identical, single-plasmid population?”

We have also clarified the negligible role of the background populations in the main text and in the legend for Figure S6B:

Line#119-122: “Following the conjugation period, both populations are plated onto a dual-antibiotic agar plate such that single transconjugant colonies are visible; dual-antibiotic selection is used to uniquely select for the plasmid-carrying population (e.g., *de novo* and adapted), while inhibiting the growth of residual parents present in the conjugation mixture.”
Line#298-299: “Finally, we assume that residual parents contribute to the total cell density of the initial population, but do not grow, consistent with our data (Fig. 6B; Supplementary Text).”

Lastly, we modified Figure 4’s captions to provide more detailed information and thus give greater clarity to our experimental setup.

Figure 4 B&C: This part of the figure does not seem to fit with the description in the main text and the legends are not detailed enough to understand what the figure parts are showing, specifically because there are different symbols used and it is not clear which ones are the simulations (I would have guessed Prediction, but then there are also symbol with error bars in B but not in C). Further, B and C seem to be switched, at least according to the main text (B should show R100-1 and C RP4).

We thank the Reviewer for pointing out this error; indeed, 4B and 4C were accidentally switched. We have corrected this so that 4B now corresponds to the R100-1 plasmid, and 4C to the RP4 plasmid. Based on the Reviewer’s suggestion, we have also updated the panels to specify “Simulation” instead of “Prediction” for clarity, and we have revised the text to better describe how replicate independent simulations were used to generate error bars on our theoretical predictions. Further, we added a legend to define symbols. We note that all error bars are included in both simulations and data panels, however some may not be visible due to their size (e.g., adapted colonies have very low standard deviation).

Briefly, these figures investigate the role of clonal heterogeneity on intrapopulation competition, as described above. To test this, we chose a plasmid without (R100-1) and with (RP4) an acquisition cost. In each case, we varied the number of initial cells to increase the amount of clonal heterogeneity present (i.e., increasing cell number increases the chances of slightly different phenotypic behavior). Our intuition was that, since R100-1 does not exhibit any lag/growth tradeoff at the clonal level, adding more clones will not increase the average growth rate. Therefore, increasing the clonal heterogeneity would be equivalent to increasing the initial cell density, causing the overall population growth rate to either remain constant or decline due to nutrient/space limitation. Conversely, since RP4 does have an acquisition cost and thus does exhibit a lag/growth tradeoff, we suspected that increasing the heterogeneity will *both* increase the initial density, and also increase the average growth rate. Thus, individual clones with faster growth rates would disproportionately dominate the population at intermediate heterogeneity (i.e., before the population becomes nutrient). In both cases, adapted cells, which do not exhibit a lag/growth tradeoff, exhibited similarly monotonic behavior. To address these comments, we have clarified the intuition for Figure 4B-C results:

Line#300-320: “Intuitively, higher initial cell densities (corresponding to additional parameter distribution sampling) simulates increasing clonal heterogeneity, and thus results in a greater chance of probabilistically selecting clones with longer lag times. In the absence of a lag/growth tradeoff (e.g., adapted transconjugants, or *de novo* transconjugants with no acquisition cost), these longer lag times are *not* correlated with higher growth rates, which remain equivalently distributed regardless of lag times. Thus, the population-level growth rate should remain constant at low to intermediate initial cell densities...In the presence of a lag/growth tradeoff, however, as initial densities increase, higher lag times are by definition coupled with (i.e., constrained by) higher growth rates. These clones are disproportionately able to dominate a population, and thus, at low to intermediate initial cell densities, the population-level growth rate should *increase* with initial density. As in the no tradeoff case, at excessively high initial cell densities growth is constrained by carrying capacity effects. Together, then, for a sufficiently costly plasmid, we expected *de novo* cells to exhibit a *biphasic relationship* where growth rates increased at low-to-intermediate initial cell densities before decreasing at high initial cell densities.”

4D: Again, the legend is not enough to understand what is shown and while the text states that intermediate acquisition costs (and therefore intermediate lag time) are favored, the right plot seems to show that intermediate lag times have the lowest percentage of plasmids?

In contrast to Figure 4B-C, which examines clonal selection within the context of a single plasmid population, simulations in 4D specifically test the outcomes of competition within a mixed, multi-plasmid population (i.e., two populations, each with a distinct plasmid). Simulations are of pairwise competitions: one population consists of an intermediate-cost plasmid that is always the same (i.e., the lag time is fixed, as indicated by the red vertical line), and the second population is variable, ranging from either shorter or longer lag times compared to the fixed intermediate-cost plasmid population (i.e., relatively lower- and higher-cost plasmids, the x-axis). The y-axis (and yellow line) reflects the outcome of this competition, and is presented as the percentage of the final population carrying the intermediate-cost plasmid. Results to the left (right) of the red line indicate the competition outcome between an intermediate-cost plasmid and one with a *lower (higher)* cost; the intersection between the yellow line and the red vertical line corresponds to the outcome of the competition where the intermediate-cost plasmid competes against itself. We note that, in all simulations, both populations were initiated at equal densities (i.e., 50% population fraction); *therefore, any final fractions >50% indicate that the intermediate-cost plasmid outcompeted (i.e., was favored over) its competitor.*

Simulations revealed that, whether or not the lag/growth tradeoff is present, an intermediate-cost plasmid will always outcompete one that has a higher cost (i.e., data to the right of the red vertical line in both panels of Figure 4D) – this is intuitive, as one would expect relatively lower

cost plasmids to do better than their higher-cost counterparts. In the absence of any lag/growth tradeoff (Figure 4D, *left*), the less costly plasmid will dominate the final population in all pairwise scenarios. Thus, intermediate-cost plasmid percentage is <50% to the left of the red vertical line and increases monotonically. However, when the tradeoff is present (Figure 4D, *right*), this is no longer the case. Instead, the intermediate-cost plasmid dominates the population *even* when competed against one that is lower-cost. Intuitively, this arises because, although the average population lag time is higher for an intermediate- compared to lower- cost plasmid, individual clones *within* the population can achieve a faster growth rate. In other words, because of the lag/growth tradeoff, individual clones belonging to an intermediate-cost plasmid population achieve an optimality between lag time and growth rate, which confers them a competitive advantage. To clarify this logic and interpretation, we have annotated the figure and the legend to better delineate our results, and updated the text to explain these simulations more clearly:

Line#327-339: “Finally, we examined the ecological dynamics at the interpopulation level; that is, how does the lag/growth tradeoff impact selection outcomes of multiple genetically distinct competing plasmids, each with their own underlying phenotypic distributions? To answer this, we considered the scenario where a fixed, intermediate-cost plasmid population (**Fig. 4D**, red line) was competed against a second plasmid-carrying population exhibiting a range of acquisition costs. We then calculated the final percentage of the intermediate-cost plasmid population, both in the absence and presence of a lag/growth tradeoff. Simulations revealed that on average, without any lag/growth tradeoff, a plasmid with a lower acquisition cost will always outcompete one with a higher acquisition cost (**Fig. 4D**, *left*). This finding is intuitive: lower-cost plasmids (i.e., shorter lag times) can dominate mixed populations earlier compared to costlier plasmids (i.e., longer lag times), when the latter has no growth advantage. In contrast, when a lag/growth tradeoff is present (i.e., a longer lag time results in a faster growth rate for individual clones), simulations predicted that intermediate-cost plasmids will outcompete those with both low and high acquisition costs (**Fig. 4D**, *right*).”

4E&F: It is confusing that high and low panels are switched in E and F (i.e., high-low in E and low-high in F).

We thank the Reviewer for raising this point and have adjusted the panels accordingly.

2. SI Table1 gives a good overview over the plasmids used, including their main characteristics, but no analysis was performed to prove or disprove any correlation between those traits (e.g. length or conjugation rate) with acquisition cost or growth rate.

We agree that it is quite interesting to correlate plasmid traits with acquisition cost and growth rate. In doing so, we found no statistical relationships between acquisition cost and any plasmid-specific traits, including the number of transconjugants, plasmid length, or adapted transconjugant growth rate ($p > 0.5$ in all cases). To address the Reviewer’s comment, we have included this analysis in Figure S2B, and updated the main text accordingly:

Line#197-202: “To further examine the generality of this observation, we first investigated whether any common plasmid features could readily explain these results. Specifically, we examined the relationship between acquisition cost and the number of transconjugants (as a proxy for the conjugation efficiency), the adapted transconjugant growth rate (as a proxy for fitness cost), and the overall plasmid size. In all cases, plasmid features were not significantly correlated with the plasmid acquisition cost ($p > 0.5$) (**Fig. S2B**).”

3. If acquisition costs mainly affects lag time, how important is that effect in an exponentially growing culture, i.e. when bacterial cells are already metabolically active, or is lag time here more thought of as the time it takes for an individual cell to start growing after it received the plasmid?

The Reviewer is correct that acquisition costs quantitate the time required for cells to metabolically adapt (including growth initiation) relative to receiving a plasmid. Indeed, different

environmental conditions and physiological states will almost certainly affect both plasmid transfer rates and acquisition cost dynamics, which we have shown previously⁹. Here, to maintain highly controlled and consistent conditions, we utilized donors and recipients harvested from stationary phase, consistent with our previous methods⁹⁻¹¹. In contrast to exponential cells that exhibit minimal/reduced lag times when transitioned to new environments, stationary cells always exhibit a lag time, and also typically have maximum heterogeneity¹²⁻¹⁶. This setup was thus ideal to characterize clonal heterogeneity associated with acquisition costs across a wide range of conditions. Expanding these results to additional metabolic/ physiological states is undoubtedly a next step we are interested in pursuing. We have modified our discussion to acknowledge this caveat, and clarify the scope:

Line#406-410: “We note that our experiments used cells initiated from stationary phase and simplified pairwise competitions. Future studies should expand the range of environmental conditions, physiological states, and population diversity studied, to assess how lag/growth outcomes may manifest in increasingly complex metabolic settings.”

4. The methods section is generally very detailed and well written, but several of the formulas in the main text and the SI are not displayed correctly (only empty squares), so they were not readable and impossible to review.

We sincerely apologize for not noticing this issue in the converted pdfs. All equations are now displayed properly.

Minor comments:

5. The second part of the introduction could be a bit more focused. It is relatively dense and mentions many concepts from molecular processes of cell metabolism to selection in clonally heterogeneous populations, which makes it a bit hard to take away which gaps and questions are addressed (as for example biomolecular processes as well as complex microbial communities are not really touched upon for the rest of the paper). Similarly, the importance of distinguishing the lag time and growth rate parts of the acquisition cost does not come out clearly.

We appreciate the feedback on improving the readability of our manuscript. We have updated the introduction to address all aspects raised by the Reviewer:

Line#68-76: “Specifically, immediately following plasmid acquisition, a cell must reallocate intracellular resources to accommodate new energetic requirements associated with plasmid DNA replication and gene expression²⁵⁻²⁹. Consequently, the magnitude and duration of the burden depends on a complex interplay between multiple interdependent biomolecular factors; for example, energy availability at the time of acquisition, plasmid-specific gene regulatory networks, and the translation order of plasmid-encoded genes, may all result in either delayed or over-production of proteins³⁰. In addition, acquiring new DNA is known to activate energetically burdensome stress responses, further exacerbating the immediate burden of a new plasmid^{31,32}.”

Line#81-87: “Further, previous population-level studies are unable to distinguish between changes in lag time, growth rate, or both. However, lag times and growth rates each uniquely contribute to various facets of bacterial dynamics, including robustness in fluctuating environments³⁴, sensitivity to stressors³⁵, and competitive success³⁶. Thus, this distinction is critical to better understand both the physiological manifestation of acquisition costs, and as a corollary, the selection dynamics at play in clonally heterogeneous populations that are most representative of those in nature³⁷⁻⁴¹.”

6. L51: ‘facilitate’ seems to be a weird word choice here

We have updated the sentence, using the word “involves” instead of facilitates (**Line#51**).

7. L85: ‘heterogenous’ populations are mentioned at several points throughout the manuscript

but it is not quite clear what they refer to and why they are important

In response to this and also point #1 above, we recognize the potential for confusion due to conflicting uses of the word heterogeneous. To fully address these comments, we have simplified the text to only use 'heterogeneity' in a single context, that is, in reference to a *clonally heterogeneous* population, or one that is genetically identical but exhibits phenotypic heterogeneity at the clonal level¹⁻⁸. In other contexts, where we were referring to heterogeneous in the context of genetically diverse/mixed population, we have modified the language to indicate it as such. In the case of this specific line, we were referring to the former; it now reads:

Line#84-87: "Thus, this distinction is critical to better understand both the physiological manifestation of acquisition costs, and as a corollary, the selection dynamics at play in clonally heterogeneous populations that are most representative of those in nature³⁷⁻⁴¹."

We have also updated several other instances of potential confusion throughout, including:

Line#146-149: "Aside from validating previous work, colony tracking also revealed extensive clonal heterogeneity within the *de novo* population; here, we use heterogeneity to denote individual colonies that belong to the same genetically identical population but exhibit phenotypic differences, as evidenced in the variability of growth curves for individual RP4 *de novo* transconjugant colonies."

Line#280-282: "Thus, we sought to assess how the lag/growth tradeoff, which is inherently dependent on underlying clonal heterogeneity, would impact selection dynamics and ecological outcomes."

Line#367-369: "However, analysis at the single-colony level revealed that these observed effects likely arise due to underlying clonal heterogeneity in the population, whereby individual clones begin growing at a wide distribution of times."

Finally, we note that the updated details in describing the (negligible) role of the background populations, as described above, should further clarify this point.

8. L181: It is unclear what instantaneous colony growth is (compared to the otherwise reported growth rates) and how it relates to 'equivalent growth rates'. Also, in Fig. S3C it looks like instantaneous growth rate is decreasing with acquisition cost for *de novo* populations – how does that fit with the trade-off between lag time and growth rate?

The manuscript refers to the maximum overall growth rate achieved as determined from either our Logistic-curve fitting method or manual fitting procedures (Methods **Line#524-536**); these rates are single parameters that describe the entire growth curve. In contrast, instantaneous growth rate calculates the change in each colony size at *every* time point, as a function of the overall average colony size. This parameter provides us with an instantaneous estimate of colony expansion, and decreases over time; this is intuitive, as the growth rate of a colony is expected to decrease until it reaches stationary phase. Measuring instantaneous growth rates allowed us to confirm that, over time, the instantaneous growth rate of *de novo* cells converge to the same as for adapted cells. This implies that observed growth trends are not an artifact of colony size, further supporting the generality of our observation. We have updated the main text, along with the Methods (**Line#538-543**) and legend of Figure S3C, accordingly:

Line#189-193: "Moreover, instantaneous colony growth rates (e.g., the change in colony size at every time point as a function of the overall average colony size) further validated these results: despite differences in colony sizes, *de novo* colonies ultimately attained the same growth rates as adapted colonies, suggesting that overall trends are not driven by size-specific artifacts of curve fitting (**Fig. S3C**)."

9. L192: For RIP113 the correlation between TTT and growth rate seems to change from neutral / decoupled to positive but maybe that is not significant?

Here our goal was to ensure acquisition phenotypes were not dependent on the antibiotic used to select for the recipient (rifampicin in all Figure 2 data). Instead, we used kanamycin for recipient selection, and measured acquisition costs for both the no-cost plasmid R100-1 and costly plasmid RIP113. We found that under both selection conditions, the following held true: (1) the magnitude of the acquisition cost did not significantly change; (2) the relationship between lag time and Time-to-threshold (TTT, a measure of the acquisition cost) remained statistically significant; (3) the relationship between growth rate and TTT remained statistically insignificant. The Reviewer referred to the observed decoupling between TTT and growth rate for RIP113 in kanamycin (Figure S4) showcasing a positive regression line; however, as the Reviewer suspected, this relationship remained statistically insignificant ($p = 0.27$). This is consistent with the insignificant, neutral relationship under rifampicin treatment (Figure 2A, $p = 0.55$). We have emphasized/clarified this point in the text:

Line#212-215: “For both plasmids, the magnitude of the acquisition cost, the statistically significant relationship between lag time and TTT, and the statistically insignificant relationship between growth rate and TTT, were maintained (Fig. S4).”

10. L215: Detrimental for plasmid acquisition or establishment? The plasmid was still acquired, it might not be able to establish in the population though.

We agree with this modification and have updated the text accordingly (**Line#238**).

11. L225: Was only mean or the whole distribution used (as the variance in TTT seems to be quite high in costly de novo populations)?

This was indeed an oversight, as the entire dataset was used (i.e., both means and standard deviations), and we have updated the text accordingly (**Line#248**).

12. L271: What do iterations correspond to here (in terms of generations)? Also, maybe give an intuition for why you are using (this) model as you do in all the other sections very nicely.

We use multiple iterations in these simulations to account for any variability that may arise from numerically-generated distributions of small sizes. Specifically, since our experiments had on average ~100 colonies, we generated distributions of the same size, which may be subject to artifacts simply due to random chance. In this way, sufficient iterations do not reflect growth generations, but rather provide confidence that the simulation results are representative of the underlying true distribution parameters. We have updated the text to clarify accordingly:

Line#295-298: “For all simulations, 50 iterations (i.e., unique parameter distributions) were averaged to ensure that predictions reflected average dynamics rather than artifacts of numerically-generated distributions of varying sizes/values.”

13. L347: Again, it is not entirely clear what is meant by heterogeneity (within a de novo plasmid population, or in a population containing de novo and adapted plasmids, or a population of recipients and donors?) and how it relates to the observed results.

We apologize for the confusion regarding heterogeneous/heterogeneity. As mentioned in response to point 7, our original use of the word heterogeneity referred to two concepts: (1) phenotypic heterogeneity as natural variation amongst individual clones that are otherwise genetically identical, and (2) heterogeneity as a result of genetically distinct populations in the

same environment. As described, we have removed all references of heterogeneous in the context of the latter. In the line cited by the Reviewer, heterogeneity referred to concept 1, namely that scanning allowed us to analyze cells at the single-colony level, which in turn revealed heterogeneity of individual colonies belonging to a single population. In addition to the edits referenced in response to point 7, we modified our text to better reflect this idea, included below. Further, we ensured that our definition of heterogeneity remained consistent throughout the revised text by only using its definition in the context of clonal heterogeneity.

Line#386-391: “Although lag time appears to drive overall acquisition dynamics, at the single-colony level, we found that clonal heterogeneity contributed significantly to population structure and ecological outcomes. Specifically, growth rate was uncorrelated with average plasmid acquisition cost at the population level. However, within a single *de novo* colony, lag time exhibited a counterintuitive tradeoff with growth rate; that is, from the same population, individual colonies with longer lag times demonstrated higher post-lag growth rates compared to those with shorter lag times.”

14. L354: The conclusion that acquisition cost might be beneficial is confusing as a tradeoff suggests that it cannot be selected against without losing beneficial growth benefits. Hence, growth benefits would be selected for, but not higher acquisition costs per se.

Respectfully, we still do believe that the acquisition cost may in fact be beneficial, although we acknowledge our explanation may have been confusing. Briefly, our results show, at the single-colony level, that populations acquiring high-cost plasmids consist of clones exhibiting longer lag times, but also potentially faster growth rates. For such plasmids, clones with higher growth rates would thus be selected for, as indicated by the Reviewer. However, these selected clones would still retain a longer lag time (i.e., a high acquisition cost). This is important when a higher cost plasmid may be competing against a lower one, as shown in Figure 4D-F: higher acquisition costs can be selected for, and even beneficial, despite exhibiting a longer lag time. This is due to the lag/growth tradeoff, which confers a growth benefit to individual cells containing higher-cost plasmids. We have clarified this reasoning in the text:

Line#396-397: “That plasmid costliness may provide bacteria with a growth benefit suggests that, unlike fitness costs, acquisition costs may never be fully selected against.”

15. Figure 1 C-E: TTT should be explained in the figure and/or the figure legend and time units should be given.

We have updated the figure and legend accordingly.

16. Figure 2: Red and green is easily the worst color choice here and as it is showing different things across the panels, those colors should not be the same. Generally, the colors in B and C do not really add additional information as they are too hard to distinguish (the only visible ones look similar to the colors used in D and E but they show different things, so that only adds confusion), so it might be better to just show them in black. Titles for the individual panels would help

We have updated the figure panels to indicate legends for individual colors, and included titles in individual panels to help interpret the results. Although we agree the colors in Figure 2B-C may be hard to distinguish, we have left them as is in the case they are useful to some readers; we have also included an adjacent legend. As suggested, we updated our color scheme in Figure 2D to avoid red/green (a huge oversight which we are grateful the Reviewer pointed out!). We also updated colors in Figure 2E to be consistent with the plasmids/conditions in 2B-C (RIP113 and RP4).

17. Figure S3A: growth rate on the y-axis is likely wrong

Thank you for pointing out this error! We have updated the y-axis to TTT.

18. Figure S6B: More explanation would be good, I assume only the growth of the *de novo* transconjugants is shown and the rest of the population was suppressed by antibiotics?

Indeed, the data shows the growth of only *de novo* transconjugants. The dual antibiotic mixture of tetracycline and rifampicin kills the pB10 donor (tetracycline-resistant and rifampicin-sensitive). Similarly, the dual antibiotic mixture kills BW25113-rif recipient (tetracycline-sensitive and rifampicin-resistant). As such, only *de novo* transconjugants with resistance to both antibiotics survive/grow. We have updated the figure legend to better describe the experiment. We have also included a more thorough methods section describing the protocol (**Line#586-597**). Finally, we have also explained the role of dual-antibiotic selection for transconjugant selection in the main text:

Line#120-122: "...dual-antibiotic selection is used to uniquely select for the plasmid-carrying population (e.g., *de novo* and adapted), while inhibiting the growth of residual parents present in the conjugation mixture."

19. S6C: There is no yellow in the figure but it is mentioned in the legend?

We have modified the color to be more apparent and updated the legend for specificity.

Reviewer #2 (Remarks to the Author):

In the manuscript "Tradeoff between lag time and growth rate drives plasmid acquisition costs", Ahmad and co-workers investigate the physiological effects associated with the acquisition of conjugative plasmids. This is an important topic as conjugation plasmids are major contributors to the dissemination of metabolic properties in bacteria, including drug resistance. Studying the effect of plasmid acquisition on the recipient strain is essential to understand the dynamics of bacterial populations undergoing conjugation and the parameters/rules of plasmid success. They use both experimental approach and mathematical modelling to compare, at the single colony level, the growth of "de novo transconjugants" (that have just acquired the plasmid) to the growth of adapted transconjugants (~24h after plasmid acquisition) (These two types of transconjugants have been defined and characterized in previous reports from the Lopatkin lab). This approach is much more powerful and informative than simply comparing plasmid-containing strains to the plasmids-free counterpart strains, and represents an obvious advantage compared to liquid culture analysis. Crucially, this work analyses growth in terms of lag time and growth rate, and a range of nine conjugative plasmids (from various incompatibility groups) are tested. The results presented provide a comprehensive description of plasmid acquisition effects at the single-transconjugant colony level (generated by single-transconjugant cells) with unprecedented amounts of details and in-depth analysis. The work demonstrates that for a given costly plasmid, transconjugant clones with longer lag time eventually exhibited a faster growth rate, suggesting the existence of a tradeoff between the adaptation period and the final growth ability. The findings are compelling (thanks to robust statistical analysis, essential controls regarding the antibiotic used for plasmid selection, validation by mathematical modelling, and the reproducibility through the plasmid range and *E. coli* or *K. p* stains), and the implication for plasmid ecology and dissemination success are well discussed. The data are well presented and the interpretations well supported by the results. However, the authors could consider discussing molecular mechanisms potentially involved in the observed heterogeneity between clones with the help of published single-cell studies. Overall, this reviewer thinks this work is an original and important addition to the field of conjugation plasmid biology, both in terms of methodology and findings.'

We are extremely grateful for the overall positive evaluation of our work, and the Reviewer's time in providing in-depth feedback. We have addressed all the concerns below and in the revised manuscript.

Comments:

1- There is no indication of overall conjugation efficiency for the different conjugative plasmids tested. Results from plating de novo transconjugants on dual-antibiotic agar plates should have given this information. The authors could consider providing conjugation efficiency estimates in the supplementary figure, if relevant and possible. This is obviously not the focus of the present work, but it still would be helpful information. This could also reveal any potential correlation between the transferability and acquisition cost of the tested plasmids. Have the authors investigated such a correlation?

The Reviewer's suggestion is quite interesting and relevant. We also have all the data to investigate these relationships further. In particular, we would like to draw the Reviewer's attention to Supplementary Table S1, which reports the number of transconjugants formed for each plasmid measured – specifically, for all experiments, conjugation is measured under precisely controlled conditions, consistent with our previous methods⁹. Critically, all donors and recipients are equilibrated at the start of each experiment such that their CFU does not differ more than ~2 fold from one another, or between experimental replicates. Therefore, the number

of transconjugants as reported in Table S1, which varies over *several orders of magnitude*, can be interpreted as a direct proxy for the conjugation efficiency comparable across all conditions in this study. Based on the Reviewer's suggestion, we used these numbers to examine any relation between conjugation efficiency (i.e., number of transconjugants) and acquisition cost. We also investigated other relevant plasmid-specific factors, including plasmid length and adapted transconjugant growth rate. In all cases, we did not find a statistically significant relationship between acquisition cost and any plasmid-specific features ($p > 0.5$ in all cases). We have included this additional analysis in Figure S2B, and a brief description in the text:

Line#197-202: "To further examine the generality of this observation, we first investigated whether any common plasmid features could readily explain these results. Specifically, we examined the relationship between acquisition cost and the number of transconjugants (as a proxy for the conjugation efficiency), the adapted transconjugant growth rate (as a proxy for fitness cost), and the overall plasmid size. In all cases, plasmid features were not significantly correlated with the plasmid acquisition cost ($p > 0.5$) (Fig. S2B)."

Although we did not find anything significant, we note this sample set is small; we are excited to continue examining such relationships and we agree conceptually with the Reviewer that, at least in some cases, these features may be related to one another.

4- L. 184. This is the first mention of "tetracycline selection". Tc selection was also probably used to generate RP4 results in Figure 1!? This should probably be stated earlier in the text. The rationale/implication of using Tc selection for the assays should also be discussed. Indeed, the experiment might or might not be possible with a bactericidal rather than a bacteriostatic antibiotic. The implication of using a translation-inhibitor (rather than a transcription-inhibitor, or DNA-damaging antibiotic, or cell-division inhibitor) might also deserve a comment.

We agree with the Reviewer that the drug class is an important decision, and we confirm that tetracycline was indeed used in Figure 1 – we have now clarified this early on in the text (**Line#138**). As suggested by the Reviewer, our initial choice of tetracycline was primarily due to its status as a bacteriostatic drug; tetracycline selection has not been associated with drug-mediated metabolic dysregulation, as is seen in response to some bactericidal antibiotics. Since the acquisition cost and resulting changes in lag time and growth rate in *de novo* transconjugants are a result of transient metabolic changes, we initially wanted to minimize such additional and potentially confounding metabolic effects. Once confirming trends under these simplified conditions, we expanded to drugs with potentially more complex secondary effects. We have updated the text accordingly:

Line#163-165: "...this initial focus on the bacteriostatic drug tetracycline is ideal, since it minimizes potential metabolic effects imposed by bactericidal antibiotics that may confound growth/lag dynamics⁴²⁻⁴⁴."

Line#220-222: "...these two antibiotics were chosen due to their distinct mechanisms of action, enzyme-based resistances, and potential for metabolic dysregulation, as compared to tetracycline⁴⁶⁻⁴⁸."

5- In the assay used, *de novo* transconjugants cells are in contact with Tc at the time of plating on dual-selection only. It follows that the host's multidrug efflux pumps (in particular AcrAB-TolC) might also play a role during plasmid establishment in the presence of antibiotics that are a substrate for these chromosomally-encoded pumps. A such role for multidrug efflux pump during plasmid acquisition under selection pressure has been demonstrated previously (PMID: 31123134) and could be considered/discussed here. This would be even more relevant than the authors did an excellent job by validating their results using other antibiotics for plasmid selection (streptomycin or carbenicillin) and different bacterial strains (*Klebsiella pneumoniae*).

We thank the Reviewer for bringing this article to our attention. Indeed, the suggested paper's results and methodology is quite interesting! These authors also observe the impact of plasmid transfer at the single colony level, especially in the context of drug resistance, noting initial over-expression of tetracycline resistance in recipient compared to donor cells, until the repressor is sufficiently produced. This connection, though quite relevant, likely only accounts for a small subset of our observations, given the range of acquisition costs that were measured under tetracycline selection, including some plasmids that exhibited no cost at all. Likewise, since these trends were maintained for multiple diverse antibiotic types encompassing various resistance mechanisms beyond efflux pumps, we do not suspect this particular mechanism to be the sole driver of the lag/growth tradeoff. We have acknowledged the relevant points in our Discussion (see excerpt below), and expanded our analysis as summarized below in response to the Reviewer's point #6:

Line#372-378: "Interestingly, recent works have found heterogeneous expression of resistance enzymes following conjugation, e.g. tetA in the presence of tetracycline^{50,51}, which may impact acquisition dynamics. Our genetic analyses did not suggest any obvious explanations for our observed tetracycline-specific differences across plasmids (e.g., both low and high costs); this, combined with consistent trends across multiple antibiotics, suggests our results are not driven by antibiotic-specific factors. However, it cannot be ruled out entirely, and examining lag/growth dynamics in the absence of antibiotic selection altogether, is an important future step."

6- L. 211-221. Regarding the heterogeneity of behaviour among individual clones, which the authors summarize as "as lag time increases, so does growth rate". The previously mention report (PMID: 31123134) also reveals the heterogeneity in the level and rate of production of the plasmid-encoded TetA efflux pump that confers resistance to tetracycline after plasmid acquisition (F in this case). In a further report, these different TetA production levels in transconjugants (corresponding to adapted transconjugants in this work) were correlated to the Tc-resistance ability at the single-cell level (PMID: 32761242). Do the authors think that the observed growth (lag time and growth rate) heterogeneity of individual clones could be related to plasmid-genes expression heterogeneity? This could help discuss molecular mechanisms underlying the behaviour revealed in this report. Can the authors think of any other molecular mechanism that could account for or contribute to the observed phenotypes?

These points are extremely interesting and important; we have reviewed both papers, and see a lot of potential synergy between the findings and our own. Indeed, we continually consider the role of antibiotic selection on observed acquisition costs. Our initial intuition was that certain drugs would make it more difficult to acquire plasmids. Based on this and the previous comment, we more closely investigated the potential role of tetracycline and tetracycline resistance. Briefly, our findings suggest that the two mechanisms are likely unrelated. Firstly, as described in response to point #1, plasmid fitness costs do not correlate with acquisition costs. This was a phenomenon we previously observed in bulk culture, and confirmed in this work at the single colony level (Figure S2B), overall suggesting that any mechanism resulting in growth defects in adapted cells is largely distinct from those in *de novo* ones. Additionally, we examined whether tet-specific genetic differences that likely lead to differences in tetA/R expression were related to observed acquisition costs under tetracycline selection. Specifically, we focused on two aspects: (1) First, we examined whether sequence similarity of the tetracycline operon (including tetA/R, promoters, and ribosome binding sites) correlated with costliness. Doing so revealed no statistically significant trends in sequence similarity that could explain phenotypic differences. (2) Second, we examined proximity of tetA to the origin of plasmid replication (oriC) and origin of transfer (oriT), under the reasonable expectation that resistance markers located more proximally to the beginning of plasmid transcription may be correlated with lower cost plasmids. As in (1), however, no such correlation was observed ($p > 0.5$ in both cases). These results confirmed that neither the sequence itself, nor its location on the plasmid, could explain

our observations. While this is not conclusive in ruling out expression differences, in total, we suspect that the effects are likely minimal. As for other molecular mechanisms, we could not agree more with the Reviewer that this is an intriguing question, and we have begun to investigate this in follow up work. As we have no conclusive results to report at this time, we have updated the Results to include this analysis, and the Discussion to speculate on further mechanisms. We hope the Reviewer agrees there is no need to expand further in the current manuscript. To address this comment, we have included the plasmid-specific analysis in **Figure S2C**, and **Table S2**; we have also described it in both the Results, along with the Discussion section as described above:

Line#202-206: “Likewise, genetic features related to tetracycline were also not predictive of acquisition cost. Specifically, neither the tetracycline operon sequence/organization, nor location of tetA on the plasmid relative to the origin of transfer (oriT) or replication (oriC), were correlated with acquisition costs (**Fig S2C**, **Table S2**). Combined, this suggests that lag time trends are independent of general plasmid, and tetracycline-specific, sequence features.”

Line#372-378: “Interestingly, recent works have found heterogeneous expression of resistance enzymes following conjugation, e.g. tetA in the presence of tetracycline^{50,51}, which may impact acquisition dynamics. Our genetic analyses did not suggest any obvious explanations for our observed tetracycline-specific differences across plasmids (e.g., both low and high costs); this, combined with consistent trends across multiple antibiotics, suggests our results are not driven by antibiotic-specific factors. However, it cannot be ruled out entirely, and examining lag/growth dynamics in the absence of antibiotic selection altogether, is an important future step.”

Line#410-413: “Further, our preliminary analysis of plasmid genetics did not reveal any obvious molecular explanations for acquisition costs; a better understanding of the underlying mechanisms would allow us to integrate experimental measurements with more bioinformatic-driven methods to investigate acquisition costs across relevant environments and clinical settings.”

Minor points:

1- All reference numbers are after full stops throughout the text, "diversity and evolution.1,2 Conjugation facilitates...". It should probably be "diversity and evolution1,2. Conjugation facilitates..."

We have revised the text accordingly.

2- In the introduction, the three following statements are justified but sound quite repetitive and rhetorically redundant. The authors might want to rephrase.

"Thus, it is critical to understand factors that facilitate plasmid dissemination to identify strategies to predict, prevent, and/or control such occurrences."

"Therefore, understanding the effects of plasmid acquisition is critical to accurately predict the overall potential and success of conjugative plasmids."

"Yet, this distinction is critical to better understand both the physiological manifestation of acquisition costs, ..."

We have updated the introduction to remove redundancy and streamline the presented logic:

Line#81-87: “Further, previous population-level studies are unable to distinguish between changes in lag time, growth rate, or both. However, lag times and growth rates each uniquely contribute to various facets of bacterial dynamics, including robustness in fluctuating environments³⁴, sensitivity to stressors³⁵, and competitive success³⁶. Thus, this distinction is critical to better understand both the physiological manifestation of acquisition costs, and as a corollary, the selection dynamics at play in clonally heterogeneous populations that are most representative of those in nature³⁷⁻⁴¹.”

3- (L. 120-121) "that single colonies are visible". At this stage, it would help to indicate "that single transconjugant colonies are visible".

We have updated the text as suggested (**Line#120**).

4- (L. 141) "more rigorous than the previous methodology". The authors might want to change "rigorous" to another word that does not undermine the rigour of previous approaches, which were also rigorous though much less precise and resolute.

We thank the Reviewer for bringing this to our attention, as our intent was certainly not to undermine the quality of any previous works. We have replaced "rigorous" with "and provides greater resolution" (**Line#142-143**).

5- Log cell density plots (Fig. 1C, Fig. S1A...) or corresponding legends should indicate the number of individual colonies monitored and plotted.

Where applicable, we have updated all figures and legends to include the number of individual colonies for single-colony-resolution work.

6- L. 164-166. "Plasmids with no or low acquisition costs (e.g., R100-1, R64drd, and pOX38)". It would help the reader to also put between brackets the list of what is interpreted as "high-cost plasmids exhibited a decoupling or positive correlation between growth rate and TTT".

We have updated the text as suggested. It now reads:

Line#173-174: "...high-cost plasmids (pB10 and pRK100) exhibited a decoupling between growth rate and TTT."

Reviewer #3 (Remarks to the Author):

Comments for Authors

The manuscript by Ahmad et al. describes a rather interesting (after reading this) and clearly understudied phenomena: the plasmid acquisition cost. Overall, the paper very clearly shows that there is a scientifically interesting “story” happening at the very moment when a bacterium receives a new genetic element. The correlation of increased lag time with the higher growth rate reveals something about the acute interaction between cells and their (sometimes) transient genetic hitchhikers. The ecological considerations of the manuscript add insights on the population and community level, and can help us understand why acquisition costs both vary and may be selected for in different ecological and environmental setups. The paper is well-written and clear. First figure is very informative and makes the paper much easier to approach. Data appears to be consistent and well-presented. Ultimately, of course, it would be interesting to know the interactions on molecular level that are responsible for the extended/decreased lag period and the subsequent growth rate. These, however, are very difficult to study. Nonetheless, this paper also contributes to the general theme on the effects that foreign genetic elements (even if relatively well adapted to that organism) cause in its new genetic environment. For example, it would be interesting to know how the acquisition costs would evolve in plasmids that are originally non-adapted to a particular host. Such changes could also reveal about the reasons behind the observed acquisition costs. Further, the manuscript suggests ways to decipher why certain plasmids are doing better than others in clinical or other relevant environment where plasmids may play a notable role. As such, clinical plasmids could be more often measured for their acquisition costs, and we could see whether there are correlations with the plasmid’s success (prevalence). Altogether, this reviewer finds no flaws in the manuscript, and it turned out to be quite surprising and insightful. As such, this reviewer would recommend the manuscript for publication, given the corrections suggested below.

We thank the Reviewer for their positive feedback on the relevance, interest, and quality of our work; we are excited to continue studying the plasmid acquisition cost at the molecular level! We have addressed all comments both in the main publication and in the text below:

Minor changes:

Line 145: “.intrapopulation heterogeneity not previously observed...” This reviewer would rather say “not previously quantified”, as most bacteriologists would know that very often any plated cells grow colonies at the same rate.

We thank the Reviewer for bringing this up; we agree and have updated the text accordingly (Line#150-151).

Methods: For the LB powder etc. just the catalogue numbers are provided without the manufacturer.

We thank the Reviewer for noting this detail. We have updated the details for the LB powder, LB agar, and all antibiotics used, to include both the manufacturer and catalogue number.

Growth curve quantification: first equation has some missing symbols.

We apologize for this oversight – all equations are now properly displayed.

References

1. Evans, C. R., Fan, Y., Weiss, K. & Ling, J. Errors during Gene Expression: Single-Cell Heterogeneity, Stress Resistance, and Microbe-Host Interactions. *mBio* **9**, e01018-18 (2018).
2. Brock, A., Chang, H. & Huang, S. Non-genetic heterogeneity — a mutation-independent driving force for the somatic evolution of tumours. *Nat Rev Genet* **10**, 336–342 (2009).
3. Geiler-Samerotte, K. *et al.* The details in the distributions: why and how to study phenotypic variability. *Current Opinion in Biotechnology* **24**, 752–759 (2013).
4. Neildez-Nguyen, T. M. A. *et al.* Epigenetic gene expression noise and phenotypic diversification of clonal cell populations. *Differentiation* **76**, 33–40 (2008).
5. Altschuler, S. J. & Wu, L. F. Cellular Heterogeneity: Do Differences Make a Difference? *Cell* **141**, 559–563 (2010).
6. Huang, S. Non-genetic heterogeneity of cells in development: more than just noise. *Development* **136**, 3853–3862 (2009).
7. Roberfroid, S., Vanderleyden, J. & Steenackers, H. Gene expression variability in clonal populations: Causes and consequences. *Critical Reviews in Microbiology* **42**, 969–984 (2016).
8. Takhaveev, V. & Heinemann, M. Metabolic heterogeneity in clonal microbial populations. *Curr Opin Microbiol* **45**, 30–38 (2018).
9. Prensky, H., Gomez-Simmonds, A., Uhlemann, A. & Lopatkin, A. J. Conjugation dynamics depend on both the plasmid acquisition cost and the fitness cost. *Molecular Systems Biology* **17**, (2021).
10. Lopatkin, A. J. *et al.* Antibiotics as a selective driver for conjugation dynamics. *Nat Microbiol* **1**, 16044 (2016).
11. Lopatkin, A. J. *et al.* Persistence and reversal of plasmid-mediated antibiotic resistance. *Nat Commun* **8**, 1689 (2017).
12. Bertrand, R. L. Lag Phase Is a Dynamic, Organized, Adaptive, and Evolvable Period That Prepares Bacteria for Cell Division. *J Bacteriol* **201**, e00697-18 (2019).
13. García-Timmermans, C. *et al.* Measuring phenotypic heterogeneity in isogenic bacterial populations using flow cytometry and Raman spectroscopy. <http://biorxiv.org/lookup/doi/10.1101/545681> (2019) doi:10.1101/545681.
14. Zhu, Y., Mustafi, M. & Weisshaar, J. C. Biophysical Properties of Escherichia coli Cytoplasm in Stationary Phase by Superresolution Fluorescence Microscopy. *mBio* **11**, e00143-20 (2020).
15. Pletnev, P., Osterman, I., Sergiev, P., Bogdanov, A. & Dontsova, O. Survival guide: Escherichia coli in the stationary phase. *Acta Naturae* **7**, 22–33 (2015).
16. Ishihama, A. Adaptation of gene expression in stationary phase bacteria. *Current Opinion in Genetics & Development* **7**, 582–588 (1997).

Reviewers' Comments:

Reviewer #1:

Remarks to the Author:

The authors have significantly improved the clarity of their manuscript and addressed all comments appropriately. As such, I recommend this study for publication.

Reviewer #2:

Remarks to the Author:

The authors have adequately addressed all my concerns and questions. I am also satisfied with the modifications made to the text and figures, which I believe improve the quality of this manuscript. Therefore, I recommend this manuscript for publication in Nature Communications.